# MEAN FIELD THEORY IN DEEP METRIC LEARNING

**Takuya Furusawa**
ZOZO Research
1-3-22 Kioicho, Chiyoda-ku, Tokyo, Japan
`takuya.furusawa@zozo.com`

## ABSTRACT

In this paper, we explore the application of mean field theory, a technique from statistical physics, to deep metric learning and address the high training complexity commonly associated with conventional metric learning loss functions. By adapting mean field theory for deep metric learning, we develop an approach to design classification-based loss functions from pair-based ones, which can be considered complementary to the proxy-based approach. Applying the mean field theory to two pair-based loss functions, we derive two new loss functions, MeanFieldContrastive and MeanFieldClassWiseMultiSimilarity losses, with reduced training complexity. We extensively evaluate these derived loss functions on three image-retrieval datasets and demonstrate that our loss functions outperform baseline methods in two out of the three datasets.

## 1    INTRODUCTION

Deep metric learning has emerged as a powerful technique for learning meaningful data representations in a variety of machine learning applications, such as image retrieval (Wang et al., 2014), face recognition (Schroff et al., 2015), and person re-idenfication (Hermans et al., 2017). The primary goal of deep metric learning is to provide an *order* to the embedding space by bringing similar instances closer and pushing dissimilar ones further apart. Typically, this is achieved by optimizing a loss function that utilizes appropriate interactions based on the distance between data points. However, conventional metric learning loss functions often suffer from high training complexity scaling polynomially with the size of the training data (Schroff et al., 2015). This challenge makes optimization of these loss functions difficult in large-scale applications and necessitates the development of sampling and mining strategies to find informative pairs.

The concept of order also plays a crucial role in statistical physics, which studies the emergent behaviors of interacting many-body systems. These many-body systems exhibit various ordered phases of matter, such as solid state, magnetism, superfluidity, and superconductivity, which cannot be predicted from their individual constituents (Anderson, 1972). While the interactions between the constituents are essential for hosting such nontrivial behaviors, they also make analyzing the systems challenging, analogous to the issue in deep metric learning.

Mean field theory (Weiss, 1907) is a powerful approach for handling the challenge associated with interacting many-body systems and provides an insightful framework for understanding their emergent behaviors. This theory introduces a mean field that represents the average behavior of constituent particles. The mean field is also known as the order parameter, as its value helps distinguish the ordered phases. The mean field theory approximates their interactions as interactions with the mean field and significantly reduces the complexity of many-body systems.

In this paper, we leverage mean field theory from statistical physics to tackle the complexity associated with deep metric learning. We develop the mean field theory for applications to loss functions in deep metric

learning and find that mean field theory is better suited for losses without anchor concepts, as opposed to the proxy-based method introduced in Movshovitz-Attias et al. (2017). In this sense, it can serve as a complementary approach to the proxy-based method for designing classification-based loss functions from pair-based ones. Furthermore, we apply the mean field theory to two pair-based loss functions and propose two new loss functions with reduced training complexity. We evaluate the proposed mean field loss functions using a benchmark protocol proposed in Musgrave et al. (2020a), which allows us a fair comparison with other baseline loss functions, and also using the traditional protocol utilized in Movshovitz-Attias et al. (2017); Kim et al. (2020). The evaluation results indicate that our mean field losses surpass other methods in two out of three image-retrieval datasets in the former protocol. Moreover, the latter evaluation protocol demonstrates that our losses not only exhibit performance improvements in three out of four image-retrieval datasets but also converge more rapidly compared to ProxyAnchor loss (Kim et al., 2020).

The main contributions of this paper are three-fold: **(1)** the introduction of the mean field theory from statistical physics as a tool to reduce the training complexity of pair-based loss functions based on an analogy between magnetism and deep metric learning; **(2)** the derivation of MeanFieldContrastive and MeanFieldClassWiseMultiSimilarity losses by application of the mean field theory to two pair-based loss functions; **(3)** the demonstration that the derived loss functions are competitive with existing baseline losses in several datasets.

## 2    RELATED WORK

### 2.1    PAIR-BASED LOSS FUNCTIONS

Pair-based loss functions, a representative category of deep metric learning losses, leverage pairwise or triplet relationships among data points. Contrastive loss (Hadsell et al., 2006) is an early example of this type, which utilizes positive and negative pairs of data and learns embeddings to place the positive pairs close and negative ones apart. Triplet loss (Weinberger & Saul, 2009) is an extension of Contrastive loss, which exploits triplets of positive, negative, and anchor data, and places an anchor embedding closer to a positive than a negative. These losses are further extended to incorporate interactions among pairs in mini-batches to improve performance and convergence speed (Sohn, 2016; Oh Song et al., 2016; Wang et al., 2019a;b). For instance, MultiSimilarity loss (Wang et al., 2019b) takes into account multiple inter-pair relationships within a mini-batch, enabling more efficient learning of the embedding space.

However, a major drawback of the pair-based losses is their sensitivity to the choice of positive and negative pairs, which is caused by the polynomial growth of the number of pairs and triplets with respect to the number of training data (Schroff et al., 2015). This usually requires sophisticated sampling and mining strategies for informative pairs to improve performance and mitigate slow convergence (Schroff et al., 2015; Shi et al., 2016; Hermans et al., 2017; Wu et al., 2017; Yuan et al., 2017; Harwood et al., 2017; Wang et al., 2019b). In this paper, we pursue an alternative approach to reduce training complexity rather than investigate these strategies. We accomplish this by leveraging the mean field theory, a concept from statistical physics.

### 2.2    CLASSFICATION-BASED LOSS FUNCTIONS

Classification-based loss functions utilize weight matrices and learn embeddings by optimizing a classification objective. Unlike pair-based losses, these losses do not face the complexity issue because they are computed in the same manner as a typical classification task.

A representative example is NormalizedSoftmax loss (Wang et al., 2017; Zhai & Wu, 2018), which is obtained by the cross-entropy loss function with an L2-normalized weight matrix. Its extensions include SphereFace (Liu et al., 2017), ArcFace (Deng et al., 2019), and CosFace (Wang et al., 2018a;b) losses obtained by modifying distance metrics and introducing margins. The losses with proxies such as ProxyTriplet and

ProxyNCA losses also belong to this category (Movshovitz-Attias et al., 2017). Such proxy losses can be derived from corresponding pair-based losses by substituting positive and negative data points with learnable embeddings called proxies while retaining anchors. ProxyAnchor loss (Kim et al., 2020) further considers interactions among samples in a mini-batch and shows promising performance in popular public datasets, surpassing other classification-based and pair-based loss functions. These losses have been extended to incorporate refined structures among data, such as graphs (Zhu et al., 2020), hierarchies (Yang et al., 2022), and others (Qian et al., 2019; Teh et al., 2020; Li et al., 2022).

In this paper, we develop the mean field theory as a technique to derive a classification-based loss from a pair-based one, addressing the challenges of the latter. Although our approach is similar to the proxy-based method, it naturally adapts to pair-based losses without anchors, which have remained unexplored by the proxy-based method.

## 3 PROPOSED APPROACH

In this section, we investigate the mean field theory and its application to deep metric learning. We first review the mean field theory for a ferromagnet in statistical mechanics by following standard statistical mechanics textbooks (e.g., see Nishimori & Ortiz (2010)). Next, based on the analogy between the ferromagnet and deep metric learning, we apply the mean field approximation to the Contrastive loss and a variant of the MultiSimilarity loss and derive classification-type loss functions with reduced training complexity.

### 3.1 MEAN FIELD THEORY FOR MAGNETS

Magnetism is one of the representative phenomena in statistical physics that show a phase transition between ordered and disordered phases. A ferromagnet is composed of a large number of microscopic magnetic spins and shows a macroscopic magnetization when a macroscopic number of the magnetic spins are aligned in the same direction.

To explain the mean field theory for a ferromagnet, we shall consider an infinite-range model[1] whose Hamiltonian (or energy) takes the following form (Nishimori & Ortiz, 2010):

$$\mathcal{H} = -\frac{J}{2N} \sum_{i,j=1}^{N} \mathbf{S}_i \cdot \mathbf{S}_j,  \tag{1}$$

with the total number of spins $N \in \mathbb{N}$ and the exchange interaction $J > 0$. Here, $\mathbf{S}_i$ represents the $i$-th constituent magnetic spin, which is regarded as a vector living on a sphere. Eq.(1) indicates that a state where spins point in the same direction is preferred energetically.[2]

According to statistical mechanics, a probabilistic distribution of the spin configuration at temperature $T$ follows the Gibbs distribution:

$$P(\{\mathbf{S}_i\}_i) = \frac{\mathrm{e}^{-\mathcal{H}/T}}{Z}, \quad Z = \int \prod_i d^2 \mathbf{S}_i \mathrm{e}^{-\mathcal{H}/T},  \tag{2}$$

and macroscopic properties of this system can be computed from the normalization factor $Z$. However, since the spins are interacting with each other, it does not look easy to compute $Z$ both analytically and numerically.

---

[1]Note that readers might worry that this model appears too simple (e.g., it lacks a notion of lattice structure). However, it is sufficient for explaining the phase transition of ferromagnets and analogous to loss functions in deep metric learning.

[2]Since the Hamiltonian (or energy) of a magnetic moment $\mathbf{m}$ in an applied magnetic field $\mathbf{B}$ is typically given by $H = -\mathbf{m} \cdot \mathbf{B}$ (e.g., Zeeman effect (Sakurai & Commins, 1995)), Eq. (1) is thought of as the interaction between a spin $\mathbf{S}_i$ and the magnetic field produced by the other spins. As a result, the interaction between spins is described by the cosine similarity, and this allows us to establish an analogy between discussions in statistical physics and deep metric learning.

To address this difficulty, we introduce the mean field theory. The central idea of the theory is to approximate the Hamiltonian (1) such that each spin interacts with an average field generated by the rest of the spins, rather than with other spins directly, thereby ignoring their fluctuations. More concretely, it means that we expand $\mathcal{H}$ with respect to fluctuations $\{(\mathbf{S}_i - \mathbf{M})\}_i$ using the identity, $\mathbf{S}_i = \mathbf{M} + (\mathbf{S}_i - \mathbf{M})$ and ignore the second-order terms of the expansion. This operation results in

$$\mathcal{H} \simeq \mathcal{H}_{\mathrm{MFT}} = \frac{JN}{2}\mathbf{M} \cdot \mathbf{M} - J\mathbf{M} \cdot \sum_{i=1}^{N} \mathbf{S}_i. \tag{3}$$

Since $\mathcal{H}_{\mathrm{MFT}}$ does not include interaction terms between spins, one can readily compute any information from the Gibbs distribution for this Hamiltonian as follows:

$$P_{\mathrm{MFT}}(\{\mathbf{S}_i\}_i) = \frac{\mathrm{e}^{-\mathcal{H}_{\mathrm{MFT}}/T}}{Z_{\mathrm{MFT}}}, \quad Z_{\mathrm{MFT}} = \int \prod_i d^2\mathbf{S}_i \mathrm{e}^{-\mathcal{H}_{\mathrm{MFT}}/T}. \tag{4}$$

The value of the mean field $\mathbf{M}$ must be determined to minimize $-\log Z_{\mathrm{MFT}}$. This condition is justified because we can show that it is equivalent to the so-called self-consistent equation

$$\mathbf{M} = \frac{1}{N} \sum_i \mathbb{E}[\mathbf{S}_i] \tag{5}$$

by differentiating $-\log Z_{\mathrm{MFT}}$ with respect to the mean field. Here, we take the expectation value over the Gibbs distribution $P_{\mathrm{MFT}}$ in Eq. (5). Since the mean field approximation is based on the expansion with respect to the fluctuations around the mean field, Eq. (5) ensures the consistency of expansion.

Overall, the mean field theory is a powerful tool that allows us to describe and analyze complex systems by approximating the interactions between individual constituents with an average field generated by the rest of the system. To draw a parallel between the above discussion and deep metric learning, let us consider the $T \to 0$ limit. In this limit, the original problem of computing $Z$ becomes one of finding a spin configuration that minimizes $\mathcal{H}$. This is analogous to a machine learning problem that seeks optimal parameters to minimize a loss function. Then, the mean field approximation reduces the problem to one of minimizing $\mathcal{H}_{\mathrm{MFT}}$ with respect to both the spins and mean field. Therefore, this observation indicates that applications of mean field theory to deep metric learning problems introduce mean fields as parameters learned to minimize their loss functions.

### 3.2 MEAN FIELD CONTRASTIVE LOSS

To study how the mean field theory works for loss functions in deep metric learning, let us begin by applying the mean field theory to Contrastive loss for the sake of simplicity and then proceed to discuss the mean field theory for a more complicated loss function.

In the following sections, we denote training data by $\mathcal{D} = \{x_i, y_i\}_{i=1}^{|\mathcal{D}|}$ composed of input data $x_i$ and its class label $y_i \in \mathcal{C} = \{1, \cdots, |\mathcal{C}|\}$. We also denote a set of data in class $c \in \mathcal{C}$ as $\mathcal{D}_c$. We extract features from the input data using a machine learning model $\mathbf{F}_\theta$, whose learnable parameters are represented by $\theta$. This model embeds the input into a manifold $\mathcal{M}$, such as $\mathbb{R}^{\mathrm{d}}$ or $S^{\mathrm{d}}$, with $\mathrm{d} \in \mathbb{N}$. We also define the distance between two embeddings, $\mathbf{F}, \mathbf{F}' \in \mathcal{M}$, as $d(\mathbf{F}, \mathbf{F}') \geq 0$. For instance, the distance can be given by the cosine distance for $\mathcal{M} = S^{\mathrm{d}}$, taking the form $d(\mathbf{F}, \mathbf{F}') = 1 - \mathbf{F} \cdot \mathbf{F}'/(||\mathbf{F}||_2||\mathbf{F}'||_2)$, or by the Euclidean distance for $\mathcal{M} = \mathbb{R}^{\mathrm{d}}$.

Contrastive loss is one of the primitive examples in deep metric learning, which is defined as

$$\begin{aligned}\mathcal{L}_{\mathrm{Cont.}} = & \frac{1}{2|\mathcal{C}|} \sum_{c \in \mathcal{C}} \frac{1}{|\mathcal{D}_c|^2} \sum_{i,j \in \mathcal{D}_c} \left[d(\mathbf{F}_\theta(x_i), \mathbf{F}_\theta(x_j)) - m_{\mathrm{P}}\right]_+ \\ & + \frac{1}{2|\mathcal{C}|} \sum_{c \neq c'} \frac{1}{|\mathcal{D}_c||\mathcal{D}_{c'}|} \sum_{i \in \mathcal{D}_c, j \in \mathcal{D}_{c'}} \left[m_{\mathrm{N}} - d(\mathbf{F}_\theta(x_i), \mathbf{F}_\theta(x_j))\right]_+,\end{aligned} \tag{6}$$

with $[x]_+ = \max(x, 0)$. Here, $m_P$ ($m_N$) is a hyperparameter that controls distances between positive (negative) instances. Note that Eq. (6) reduces to the Hamiltonian (1) when $|\mathcal{C}| = 1$, $m_P < 0$, and $\mathcal{M} = S^2$, and it requires the $\mathcal{O}(|\mathcal{D}|^2)$ training complexity as it is parallel to the situation in Sec. 3.1.

This analogy encourages us to apply the mean field theory in order to obtain a simpler loss function. Since we have multiple classes here, we shall introduce mean fields $\{\mathbf{M}_c\}_{c \in \mathcal{C}}$ and expand $\mathcal{L}_{\text{Cont.}}$ with respect to fluctuations around them. Note that, in contrast to the single-class case, we must impose the following conditions to constrain relative distance among the mean fields:

$$\left[ m_N - d(\mathbf{M}_c, \mathbf{M}_{c'}) \right]_+ = 0 \quad \left( c \neq c' \right). \tag{7}$$

This condition means that we should explore configurations of the mean fields which minimize $\mathcal{L}_{\text{Cont.}}$ at the zeroth order of expansions around the mean fields. In practice, we take into account these constraints softly.

In the expansion around the mean fields, we ignore all cross-product terms of the fluctuations keeping any others so that we reduce the complexity while taking into account the higher-order terms of self-interactions. By summing over the remaining terms, we obtain MeanFieldContrastive (MFCont.) loss, which takes the following form:

$$\mathcal{L}_{\text{MFCont.}} = \frac{1}{|\mathcal{C}|} \sum_{c \in \mathcal{C}} \frac{1}{|\mathcal{D}_c|} \sum_{i \in \mathcal{D}_c} \left( \left[ d(\mathbf{F}_\theta(x_i), \mathbf{M}_c) - m_P \right]_+ + \sum_{c'(\neq c)} \left[ m_N - d(\mathbf{F}_\theta(x_i), \mathbf{M}_c) \right]_+ \right)$$
$$+ \frac{\lambda_{\text{MF}}}{|\mathcal{C}|} \sum_{c \neq c'} \left[ m_N - d(\mathbf{M}_c, \mathbf{M}_{c'}) \right]_+^2, \tag{8}$$

where we impose the constraints (7) softly by $\lambda_{\text{MF}} > 0$. Note that resummation here naïvely produces unstable terms, $\{-[m_N - (d(\mathbf{M}_c, \mathbf{M}_{c'})]_+\}_{c,c'}$, but they vanish, thanks to the constraints (7).[3] We emphasize that we must minimize $\mathcal{L}_{\text{MFCont.}}$ by optimizing both $\mathbf{M}_c$ and $\theta$, and we can readily show that the optimal mean fields satisfy $\mathbf{M}_c = \sum_{i \in \mathcal{D}_c} \mathbf{F}_\theta(x_i)/|\mathcal{D}_c|$ at the first order of fluctuations. Note that this equation is inherently satisfied by the optimal solution. This point should be contrasted to the center loss (Wen et al., 2016), which necessitates the updating of class centers in every batch. Furthermore, in contrast to the proxy-based method, which can be applied only to a pair-based loss with an anchor, the mean field theory is applicable to wider types of pair-based loss functions as it is based on the Taylor expansions.

### 3.3 MEAN FIELD CLASS-WISE MULTISIMILARITY LOSS

Lastly, we consider the mean field approximation of a loss function which incorporates interactions within a mini-batch similar to MultiSimilarity (Wang et al., 2019b) and ProxyAnchor (Kim et al., 2020) losses. However, the mean field approximation relies on expansions around mean fields, and thus, a loss function symmetric with respect to $x_i$ and $x_j$ (i.e., without anchors) would be more desirable for our purpose. (See the supplement for the application to a loss with an anchor.) Since most loss functions do not exhibit such a symmetric property, we propose the following loss function that satisfies these requirements:

$$\mathcal{L}_{\text{CWMS}} = \frac{1}{\alpha|\mathcal{C}|} \sum_{c \in \mathcal{C}} \log \left[ 1 + \frac{\sum_{i,j \in \mathcal{D}_c, i \neq j} e^{\alpha(d(\mathbf{F}_\theta(x_i), \mathbf{F}_\theta(x_j)) - \delta)}}{2|\mathcal{D}_c|^2} \right]$$
$$+ \frac{1}{2\beta|\mathcal{C}|} \sum_{c \neq c'} \log \left[ 1 + \frac{\sum_{i \in \mathcal{D}_c, j \in \mathcal{D}_{c'}} e^{-\beta(d(\mathbf{F}_\theta(x_i), \mathbf{F}_\theta(x_j)) - \delta)}}{|\mathcal{D}_c||\mathcal{D}_{c'}|} \right], \tag{9}$$

---

[3]Here, 'resummation' refers to the process of transforming an infinite series back into a function using a method such as Taylor expansion. An 'unstable term' is a term that could potentially violate the positivity of a loss function.

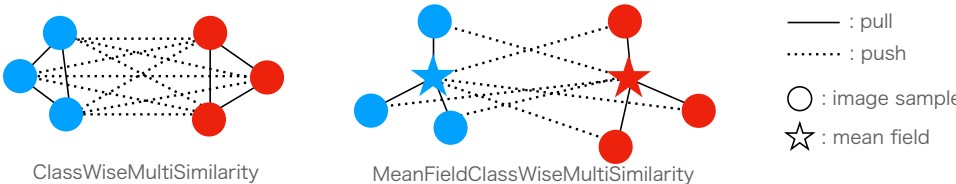

Figure 1: Schematic illustration of interactions in CWMS (left) and MFCWMS (right) losses. Each color indicates a class to which an embedding and a mean field belong.

with hyperparameters $\alpha > 0$, $\beta > 0$, and $\delta \in [-1, 1]$. Since Eq. (9) takes a similar form to MultiSimilarity loss (Wang et al., 2019b) but incorporates interactions among negative samples in a class-wise manner, we refer to it as ClassWiseMultiSimilarity (CWMS) loss.

Next, we derive the mean field counterpart of this loss function. Here, the logits in the first and second terms take forms similar to the positive and negative interactions found in Contrastive loss (6). Repeating the discussion in Sec. 3.2, we derive MeanFieldClassWiseMultiSimilarity (MFCWMS) loss:

$$
\begin{aligned}
\mathcal{L}_{\text{MFCWMS}} =& \frac{1}{\alpha|\mathcal{C}|} \sum_{c \in \mathcal{C}} \log \left[ 1 + \frac{\sum_{i \in \mathcal{D}_c} e^{\alpha(d(\mathbf{F}_\theta(x_i), \mathbf{M}_c) - \delta)}}{|\mathcal{D}_c|} \right] \\
&+ \frac{1}{2\beta|\mathcal{C}|} \sum_{c \neq c'} \log \left[ 1 + \frac{\sum_{i \in \mathcal{D}_c} e^{-\beta(d(\mathbf{F}_\theta(x_i), \mathbf{M}_{c'}) - \delta)}}{|\mathcal{D}_c|} + \frac{\sum_{j \in \mathcal{D}_{c'}} e^{-\beta(d(\mathbf{M}_c, \mathbf{F}_\theta(x_j)) - \delta)}}{|\mathcal{D}_{c'}|} \right] \\
&+ \frac{\lambda_{\text{MF}}}{|\mathcal{C}|} \sum_{c \neq c'} \left( \log \left[ 1 + e^{-\beta(d(\mathbf{M}_c, \mathbf{M}_{c'}) - \delta)} \right] \right)^2,
\end{aligned}
\tag{10}
$$

where we also introduce the soft constraint for the mean fields to ignore unstable terms produced in the resummation.[4] Compared to MeanFieldContrastive loss, this loss function incorporates interactions of positive samples as well as those of negative ones in a class-wise manner like ProxyAnchor loss.

## 4 EXPERIMENTS

Let us see the effectiveness of the proposed mean field losses by evaluating their image-retrieval performance in several public datasets. We employ the recently proposed benchmarking scheme (Musgrave et al., 2020a) as well as the traditional one used in Movshovitz-Attias et al. (2017); Kim et al. (2020). We compare our mean field losses and existing loss functions, such as MultiSimilarity and ProxyAnchor losses. We also explore the effect of hyperparameters on their evaluation metrics. In our experiment, we use precision at 1 (P@1), R-precision (RP), and mean average precision at R (MAP@R) as evaluation metrics. In particular, we focus on MAP@R in the main paper (see the supplement for P@1 and RP) because it reflects the correctness of the ranking for retrievals and is a suitable metric to evaluate the quality of the embedding space (Musgrave et al., 2020a). Note that we implement our experiments in PyTorch (Paszke et al., 2019) and utilize the PyTorch Metric Learning library (Musgrave et al., 2020b) to implement baseline models.

---

[4]Rigorously speaking, we cannot minimize $\{e^{-\beta(d(\mathbf{M}_c, \mathbf{M}_{c'}) - \delta)}\}_{c,c'}$ simultaneously. However, focusing on the region with $\beta \gg 1$, we can easily find mean field configurations satisfying $e^{-\beta(d(\mathbf{M}_c, \mathbf{M}_{c'}) - \delta)} \ll 1$ at the same time. This is enough to ignore unstable terms in practice.

## 4.1 DATASETS

In our experiments, we utilize four publicly available image-retrieval datasets, CUB-200-2011 (CUB) (Wah et al., 2011), Cars-196 (Cars) (Krause et al., 2013), Stanford Online Products (SOP) (Oh Song et al., 2016), and InShop (Liu et al., 2016). CUB comprises 11,788 images of birds categorized into 200 classes, while the Cars dataset consists of 16,185 images of 196 car classes. In CUB, the first 100 classes (5,864 images) are used for the training dataset, and the remaining 100 classes (5,924 images) are allocated for the test dataset. Similarly, Cars was split into 8,054 training images (98 classes) and 8,131 test images (98 classes). The SOP dataset contains 22,634 classes with 120,053 product images. The initial 11,318 classes with 59,551 images are used for training, and the remaining 11,316 classes with 60,502 images are allocated for testing. Lastly, the InShop dataset features 52,712 images of 7,982 fashion products, with 25,882 images from 3,997 classes used for training and 26,830 images from 3,985 classes allocated for testing, which are further divided into query (14,218 images) and gallery (12,612 images) subsets.

## 4.2 IMPLEMENTATION DETAILS

As a backbone embedding model $\mathbf{F}_\theta(x)$, we employ the inception network with batch normalization (BN-Inception) (Ioffe & Szegedy, 2015), which is pretrained for the classification task on the ImageNet dataset (Russakovsky et al., 2015). We reduce the embedding dimensions by inserting a fully-connected layer with ReLU activation functions in the first scheme and replacing its last linear layer with that of desired dimensions in the second scheme. In both cases, we apply random resized cropping and random horizontal flipping to all inputs during training and only center cropping during evaluation.

In the modern benchmarking protocol, we perform 50 iterations of Bayesian optimization for hyperparameters in loss functions including the learning rate for proxies and mean fields for a fair comparison. We split a dataset into train–valid (the first half classes) and test datasets (the remaining). The train–valid set was further divided into four partitions in a class disjoint manner, and we performed four-fold cross-validation based on the leave-one-out method in each iteration. In each cross-validation step, we train a model with embedding dimensions set to 128 and batch size set to 32 until MAP@R for the validation data converges. The Bayesian optimization aims to maximize the average of the four validation metrics. Note that we sample images so that each mini-batch is composed of 32 classes (8 classes) and 1 image (4 images) per class for classification-based (pair-based) losses, and we utilize the RMSprop optimizer with learning rate $10^{-6}$ for the embedding model. In the test stage, we perform cross-validation again with the best hyperparameters, resulting in four embedding models. Using these models, we evaluate performance on the test dataset in the following two different ways: mean of the metrics computed from the 128-dimensional (128D) embeddings (separated) and those from 512D embeddings made of the four 128D ones (concatenated). We repeat this observation 10 times and report their average values with 95% intervals. We carry out the experiments on a single NVIDIA V100 GPU.

In contrast, in the traditional evaluation protocol, we use the predefined train–test splits described in Sec. 4.1 and train a model for up to 60 epochs with embedding dimensions 512 and batch size 128, setting the patience for early stopping to 5 to accelerate the experiments. In this case, we use AdamW optimizer (Loshchilov & Hutter, 2017) with the learning rate $10^{-4}$ for the embedding model, setting the learning rate for proxies to $10^{-2}$ and that for mean fields to $2 \times 10^{-1}$. The hyperparameters for ProxyAnchor loss are fixed to $(\alpha, \delta) = (32, 10^{-1})$, while we set $(m_\text{P}, m_\text{N}, \lambda_\text{MF}) = (0.02, 0.3, 0)$ for MFCont. loss and $(\alpha, \beta, \delta, \lambda_\text{MF}) = (0.01, 80, 0.8, 0)$ for MFCWMS loss in default. We chose these default parameters according to the results of the Bayesian optimization and the discussion in Sec. 4.4 and the supplement. We repeat the above procedure 10 times and report the averages of the metrics computed from the test embedding with the best MAP@R with 95% confidence intervals. The experiments on the CUB and Cars (SOP and InShop) datasets are carried out on a single NVIDIA V100 (A100) GPU. Note that we also present the results of the traditional evaluation protocol with VisionTransformer (Dosovitskiy et al., 2020) in the supplement.

Table 1: MAP@R obtained from the modern protocol in CUB, Cars, and SOP. We carry out test runs 10 times and present the averaged metrics along with their confidence intervals. The best result within each block is underlined, while the overall best results for all losses are highlighted in bold. ProxyAnchor loss failed to converge in SOP in our settings. See the supplement for complete results.

| | CUB | | Cars | | SOP | |
|---|---|---|---|---|---|---|
| **Loss** | **128D** | **512D** | **128D** | **512D** | **128D** | **512D** |
| ArcFace | $21.5 \pm 0.1$ | $26.4 \pm 0.2$ | $18.3 \pm 0.1$ | $\underline{27.6 \pm 0.1}$ | $41.5 \pm 0.2$ | $\underline{47.4 \pm 0.2}$ |
| CosFace | $21.2 \pm 0.2$ | $\underline{26.5 \pm 0.3}$ | $18.5 \pm 0.1$ | $27.0 \pm 0.3$ | $41.0 \pm 0.2$ | $46.8 \pm 0.2$ |
| MS | $21.0 \pm 0.2$ | $26.2 \pm 0.2$ | $18.7 \pm 0.3$ | $27.2 \pm 0.4$ | $41.9 \pm 0.2$ | $46.7 \pm 0.2$ |
| MS+Miner | $20.8 \pm 0.2$ | $25.9 \pm 0.2$ | $18.5 \pm 0.2$ | $26.9 \pm 0.4$ | $41.9 \pm 0.3$ | $46.6 \pm 0.3$ |
| ProxyNCA | $18.8 \pm 0.2$ | $23.8 \pm 0.2$ | $17.4 \pm 0.1$ | $26.8 \pm 0.2$ | $\underline{42.7 \pm 0.1}$ | $46.7 \pm 0.1$ |
| ProxyAnch. | $21.7 \pm 0.2$ | $\underline{26.5 \pm 0.2}$ | $\mathbf{19.4 \pm 0.2}$ | $26.8 \pm 0.3$ | $-$ | $-$ |
| Cont. | $21.0 \pm 0.1$ | $26.4 \pm 0.2$ | $17.0 \pm 0.3$ | $24.9 \pm 0.5$ | $41.1 \pm 0.2$ | $45.3 \pm 0.2$ |
| MFCont. | $\underline{22.0 \pm 0.1}$ | $\mathbf{27.2 \pm 0.1}$ | $\underline{18.1 \pm 0.1}$ | $\underline{27.4 \pm 0.2}$ | $\underline{43.6 \pm 0.4}$ | $\underline{47.0 \pm 0.2}$ |
| CWMS | $21.5 \pm 0.3$ | $26.9 \pm 0.3$ | $\underline{19.3 \pm 0.3}$ | $\mathbf{27.8 \pm 0.3}$ | $41.5 \pm 0.2$ | $45.1 \pm 0.2$ |
| MFCWMS | $\mathbf{22.1 \pm 0.1}$ | $\underline{27.0 \pm 0.1}$ | $18.9 \pm 0.2$ | $\underline{27.0 \pm 0.3}$ | $\mathbf{44.6 \pm 0.2}$ | $\mathbf{48.3 \pm 0.2}$ |

Table 2: MAP@R values and epochs with the best accuracies obtained using the traditional protocol on the CUB, Cars, SOP, and InShop datasets. The best result in each column is underlined.

| | CUB | | Cars | | SOP | | InShop | |
|---|---|---|---|---|---|---|---|---|
| **Loss** | **MAP@R** | **Epoch** | **MAP@R** | **Epoch** | **MAP@R** | **Epoch** | **MAP@R** | **Epoch** |
| ProxyAnch. | $25.1 \pm 0.2$ | $11.5 \pm 1.4$ | $\underline{26.3 \pm 0.2}$ | $23.6 \pm 1.8$ | $51.5 \pm 0.3$ | $40.5 \pm 6.5$ | $65.5 \pm 0.1$ | $31.3 \pm 7.2$ |
| MFCont. | $\underline{25.3 \pm 0.3}$ | $\underline{4.4 \pm 0.4}$ | $24.7 \pm 0.2$ | $10.1 \pm 0.5$ | $\underline{52.9 \pm 0.1}$ | $25.1 \pm 1.4$ | $\underline{67.7 \pm 0.2}$ | $24.1 \pm 2.7$ |
| MFCWMS | $\underline{25.3 \pm 0.3}$ | $4.7 \pm 0.6$ | $24.0 \pm 0.2$ | $\underline{8.5 \pm 0.8}$ | $52.7 \pm 0.0$ | $\underline{23.0 \pm 1.3}$ | $67.5 \pm 0.4$ | $\underline{20.6 \pm 2.1}$ |

## 4.3 BENCHMARK RESULTS

Based on the first protocol, we study the performance of our loss functions on three datasets; CUB, Cars, and SOP. We compare our loss functions with existing ones, such as Contrastive, MultiSimilarity, ArcFace, CosFace, ProxyNCA, and ProxyAnchor losses. Note that recently proposed losses with additional structures (Zheng et al., 2021; Ko et al., 2021; Deng & Zhang, 2022; Yang et al., 2022; Li et al., 2022) are not included as they are out of our focus. The experimental results are summarized in Table 1 (see the supplement for complete results). First, the mean field losses show better performance than their original pair-based losses in most cases, indicating that applying mean field theory not only reduces training complexity but also results in better embeddings. This is perhaps because the mean field losses can reduce the noise introduced in pairwise comparisons by the mean fields. Furthermore, the mean field losses consistently outperform other baseline methods in both separate and concatenated MAP@R for the CUB and SOP datasets. However, in Cars, ProxyAnchor and CWMS losses show better performance than the mean field losses, which might imply the importance of interactions within batch samples in this dataset.

We also test the performance in the four datasets described in Sec. 4.1 following the traditional protocol. As shown in Table 2, our mean field losses outperform ProxyAnchor loss in MAP@R except for the Cars dataset, which is consistent with the first experiment. The improvement in accuracy is evident in the larger datasets. Besides, MFCont. and MFCWMS losses converge faster than ProxyAnchor loss in all the datasets.

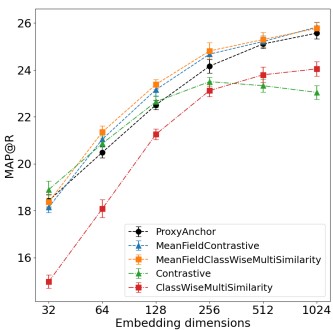
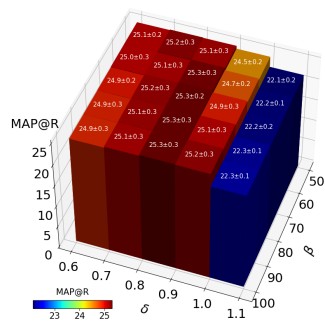

Figure 2: MAP@R versus embedding dimensions in the CUB dataset.

Figure 3: MAP@R against $\beta$ and $\delta$ of MFCWMS loss in the CUB dataset.

### 4.4 IMPACT OF HYPERPARAMETERS

**Embedding dimensions.** Since embedding dimensions are crucial hyperparameters controlling the performance of image retrieval, we investigate their effect on the accuracy (MAP@R). On the CUB dataset, we run the traditional experiments for ProxyAnchor, Cont., CWMS, MFCont., and MFCWMS losses by varying the embedding dimensions from 32 to 1024. The results are shown in Fig. 2. We find that our mean field losses monotonically increase their performance and show better performance than the baselines in most cases. Compared to ProxyAnchor loss, the improvement is larger in relatively small dimensions (64, 128, and 256).

$\beta$ **and** $\delta$ **of MFCWMS.** We also explore the effect of $\beta$ and $\delta$ of MFCWMS loss in the CUB dataset. We varied $\beta$ from 50 to 90 and $\delta$ from 0.6 to 1, fixing $(\alpha, \lambda_{\mathrm{MF}})$ to $(0.01, 0)$ and computed the MAP@R for the test data. The results are summarized in Fig. 3. Fig. 3 ensures the competitive performance of the MFCWMS loss is stable against a choice of the hyperparameters. The preferred $\beta$ gradually decreases as $\delta$ decreases.

## 5 CONCLUSION

In this paper, we applied the mean field theory in statistical physics to Contrastive loss and ClassWiseMultiSimilarity loss, a variant of MultiSimilarity loss (Wang et al., 2019b) without anchors, and derived MeanFieldContrastive and MeanFieldClassWiseMultiSimilarity losses. We extensively evaluated the proposed loss functions and compared them with the existing baseline methods using both modern and traditional benchmark protocols. The evaluation results demonstrate that the proposed loss functions outperform the baselines in the CUB and SOP datasets in the former protocol, and in the CUB, SOP, and InShop datasets in the latter. These findings highlight the potential of mean field theory as a powerful tool for simplifying and improving deep metric learning performance in various machine learning applications.

In future work, it would be worthwhile to explore applications of the proposed approach to deep metric learning in the multi-label setting (Kobayashi, 2023). Furthermore, since the mean field theory was originally introduced to elucidate the phase transition and scaling laws in ferromagnets, it would be also interesing to apply the mean field theory to explore the phase diagram of deep metric learning.

ACKNOWLEDGMENTS

We thank Yuki Saito, Ryosuke Goto, Masanari Kimura and Yuki Hirakawa for their useful comments on our manuscript.

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

## A DETAILS OF MEAN FIELD THEORY

### A.1 DERIVATION OF EQ. (5)

Let us show the equivalence between minimizing $-\log Z_{\mathrm{MFT}}$ and the self-consistent equation (5). By differentiating $-\log Z_{\mathrm{MFT}}$ with respect to the mean field, we get

$$
\begin{aligned}
-\frac{\partial}{\partial \mathbf{M}} \log Z_{\mathrm{MFT}} &= -\frac{\partial}{\partial \mathbf{M}}\Big[-\frac{JN}{2T}\mathbf{M}\cdot\mathbf{M} + \sum_{i=1}^{N}\log\int d^2\mathbf{S}_i \mathrm{e}^{\frac{J}{T}\mathbf{M}\cdot\mathbf{S}_i}\Big] \\
&= -\frac{JN}{T}\Big[-\mathbf{M} + \frac{1}{N}\sum_{i=1}^{N}\frac{\int d^2\mathbf{S}_i \mathrm{e}^{\frac{J}{T}\mathbf{M}\cdot\mathbf{S}_i}\mathbf{S}_i}{\int d^2\mathbf{S}_i \mathrm{e}^{\frac{J}{T}\mathbf{M}\cdot\mathbf{S}_i}}\Big] \\
&= -\frac{JN^2}{T}\Big[-\mathbf{M} + \frac{1}{N}\sum_{i=1}^{N}\mathbb{E}[\mathbf{S}_i]\Big],
\end{aligned}
\tag{11}
$$

and thus, $-\partial\log Z_{\mathrm{MFT}}/\partial\mathbf{M} = 0$ is equivalent to Eq. (5).

### A.2 SOLVING THE INFINITE-RANGE MODEL

For the infinite-range model (1), we can show that the mean field approximation becomes exact in the $N\to\infty$ limit. To see this, we shall compute $-\log Z$ as follows:

$$
\begin{aligned}
-\log Z &= -\log\int\prod_i d^2\mathbf{S}_i\mathrm{e}^{\frac{J}{2NT}\sum_{i,j}\mathbf{S}_i\cdot\mathbf{S}_j} = -\log\int\prod_i d^2\mathbf{S}_i\mathrm{e}^{\frac{J}{2NT}\sum_i\mathbf{S}_i\cdot\sum_j\mathbf{S}_j} \\
&= -\log\Big[\int\prod_i d^2\mathbf{S}_i\mathrm{e}^{\frac{J}{2NT}\sum_i\mathbf{S}_i\cdot\sum_j\mathbf{S}_j}\frac{1}{\sqrt{(2\pi T/(JN))^3}}\int d^3\mathbf{M}\mathrm{e}^{-\frac{JN}{2T}\mathbf{M}\cdot\mathbf{M}}\Big] \\
&= -\log\Big[\int d^3\mathbf{M}\int\prod_i d^2\mathbf{S}_i\mathrm{e}^{\frac{J}{2NT}\sum_i\mathbf{S}_i\cdot\sum_j\mathbf{S}_j}\mathrm{e}^{-\frac{JN}{2T}\mathbf{M}\cdot\mathbf{M}}\Big] + \log\sqrt{(2\pi T/(JN))^3} \\
&= -\log\Big[\int d^3\mathbf{M}\int\prod_i d^2\mathbf{S}_i\mathrm{e}^{\frac{J}{2NT}\sum_i\mathbf{S}_i\cdot\sum_j\mathbf{S}_j}\mathrm{e}^{-\frac{JN}{2T}(\mathbf{M}-\frac{\sum_i\mathbf{S}_i}{N})\cdot(\mathbf{M}-\frac{\sum_i\mathbf{S}_i}{N})}\Big] + \log\sqrt{(2\pi T/(JN))^3} \\
&= -\log\Big[\int d^3\mathbf{M}\int\prod_i d^2\mathbf{S}_i\mathrm{e}^{-\frac{JN}{2T}\mathbf{M}\cdot\mathbf{M}+\frac{J}{T}\mathbf{M}\cdot\sum_i\mathbf{S}_i}\Big] + \log\sqrt{(2\pi T/(JN))^3},
\end{aligned}
\tag{12}
$$

where we introduced a Gaussian integration in the second line and shifted the integration variable $\mathbf{M}$ by a vector $\sum_i\mathbf{S}_i/N$. Note that this trick is known as the Hubbard–Stratonovich transformation in statistical physics (Hubbard, 1959; Stratonovich, 1957). In the last line, the exponent of the integrand is equal to the mean field Hamiltonian (3), but the above equation differs from $-\log Z_{\mathrm{MFT}}$ by the integration over $\mathbf{M}$.

However, the integration over $\mathbf{M}$ can be replaced by minimization with respect to $\mathbf{M}$ in the $N\to\infty$ limit. To see this, we first introduce

$$
V(\frac{J||\mathbf{M}||_2}{T}) = -\log\int d^2\mathbf{S}\mathrm{e}^{\frac{J}{T}\mathbf{M}\cdot\mathbf{S}},
\tag{13}
$$

where the right-hand side is independent of the direction of $\mathbf{M}$ because the integration of $\mathbf{S}$ is performed over a sphere. Then, we find

$$
\int d^3\mathbf{M}\int\prod_i d^2\mathbf{S}_i\mathrm{e}^{-\frac{JN}{2T}\mathbf{M}\cdot\mathbf{M}+\frac{J}{T}\mathbf{M}\cdot\sum_i\mathbf{S}_i} = \int d^3\mathbf{M}\mathrm{e}^{-N\big[\frac{J}{2T}\mathbf{M}\cdot\mathbf{M}+V(\frac{J||\mathbf{M}||_2}{T})\big]}.
\tag{14}
$$

Taking $N\to\infty$, we see that the most contribution comes from $\mathbf{M}$ minimizing the integrand. Thus, Eq. (12) reduces $-\log Z_{\mathrm{MFT}}$ in this limit.

### A.3 DERIVATION OF MFCONT. AND MFCWMS LOSSES

To understand the derivation of MFCont. and MFCWMS losses, let us consider the expansion and resummation of $\phi(d(\mathbf{F}_\theta(x_i),\mathbf{F}_\theta(x_j)))$ with an arbitrary function $\phi:\mathbb{R}\to\mathbb{R}$ and $x_i\in\mathcal{D}_c, x_j\in\mathcal{D}_{c'}$. We expand this function by $(\mathbf{M}_c-\mathbf{F}_\theta(x_i))$

and $(\mathbf{M}_c - \mathbf{F}_\theta(x_j))$, ignoring their cross terms, and we obtain

$$
\begin{aligned}
\phi(d(\mathbf{F}_\theta(x_i), \mathbf{F}_\theta(x_j))) \simeq & \phi(d(\mathbf{M}_c, \mathbf{M}_{c'})) + \sum_{n=1}^{\infty} \sum_{k_1, \cdots, k_n} \frac{\partial^n \phi(d(\mathbf{M}_c, \mathbf{M}_{c'}))}{\partial M_{c,k_1} \cdots \partial M_{c,k_n}} (\mathbf{M}_c - \mathbf{F}_\theta(x_i))_{k_1} \cdots (\mathbf{M}_c - \mathbf{F}_\theta(x_i))_{k_n} \\
& + \sum_{n=1}^{\infty} \sum_{k_1, \cdots, k_n} \frac{\partial^n \phi(d(\mathbf{M}_c, \mathbf{M}_{c'}))}{\partial M_{c',k_1} \cdots \partial M_{c',k_n}} (\mathbf{M}_{c'} - \mathbf{F}_\theta(x_j))_{k_1} \cdots (\mathbf{M}_{c'} - \mathbf{F}_\theta(x_j))_{k_n} \\
= & -\phi(d(\mathbf{M}_c, \mathbf{M}_{c'})) + \sum_{n=0}^{\infty} \sum_{k_1, \cdots, k_n} \frac{\partial^n \phi(d(\mathbf{M}_c, \mathbf{M}_{c'}))}{\partial M_{c,k_1} \cdots \partial M_{c,k_n}} (\mathbf{M}_c - \mathbf{F}_\theta(x_i))_{k_1} \cdots (\mathbf{M}_c - \mathbf{F}_\theta(x_i))_{k_n} \\
& + \sum_{n=0}^{\infty} \sum_{k_1, \cdots, k_n} \frac{\partial^n \phi(d(\mathbf{M}_c, \mathbf{M}_{c'}))}{\partial M_{c',k_1} \cdots \partial M_{c',k_n}} (\mathbf{M}_{c'} - \mathbf{F}_\theta(x_j))_{k_1} \cdots (\mathbf{M}_{c'} - \mathbf{F}_\theta(x_j))_{k_n} \\
= & -\phi(d(\mathbf{M}_c, \mathbf{M}_{c'})) + \phi(d(\mathbf{F}_\theta(x_i), \mathbf{M}_{c'})) + \phi(d(\mathbf{M}_c, \mathbf{F}_\theta(x_j))).
\end{aligned}
\tag{15}
$$

Here, the resummation induces the unstable term $-\phi(d(\mathbf{M}_c, \mathbf{M}_{c'}))$ for $c \neq c'$, which could result in a non-convex loss function, but it vanishes thanks to the constraint (7). Note that for $c = c'$, the unstable term becomes a constant since $d(\mathbf{M}_c, \mathbf{M}_c) = 0$.

In the case of MFCont., $\phi(d)$ is given by $[d - m_P]_+$ or $[m_N - d]_+$, while it is given by $e^{\alpha[d-\delta]}$ or $e^{-\beta[d-\delta]}$ in the case of MFCWMS. For instance, the first and second terms in MFCWMS loss (10) are obtained as follows:

$$
\begin{aligned}
\sum_c \log \left[ 1 + \frac{\sum_{i,j \in \mathcal{D}_c, i \neq j} e^{\alpha(d(\mathbf{F}_\theta(x_i), \mathbf{F}_\theta(x_j)) - \delta)}}{2|\mathcal{D}_c|^2} \right] & = \sum_c \log \left[ 1 + \frac{\sum_{i,j \in \mathcal{D}_c} e^{\alpha(d(\mathbf{F}_\theta(x_i), \mathbf{F}_\theta(x_j)) - \delta)}}{2|\mathcal{D}_c|^2} - \frac{e^{\alpha(1-\delta)}}{2|D_c|} \right] \\
& \simeq \sum_c \log \left[ 1 + \frac{\sum_{i \in \mathcal{D}_c} e^{\alpha(d(\mathbf{F}_\theta(x_i), \mathbf{M}_c)) - \delta)}}{|\mathcal{D}_c|} \right],
\end{aligned}
\tag{16}
$$

and

$$
\begin{aligned}
& \sum_{c \neq c'} \log \left[ 1 + \frac{\sum_{i \in \mathcal{D}_c, j \in \mathcal{D}_{c'}} e^{-\beta(d(\mathbf{F}_\theta(x_i), \mathbf{F}_\theta(x_j)) - \delta)}}{|\mathcal{D}_c||\mathcal{D}_{c'}|} \right] \\
& = \sum_{c \neq c'} \log \left[ 1 + \frac{\sum_{i \in \mathcal{D}_c} e^{-\beta(d(\mathbf{F}_\theta(x_i), \mathbf{M}_{c'}) - \delta)}}{|\mathcal{D}_c|} + \frac{\sum_{j \in \mathcal{D}_{c'}} e^{-\beta(d(\mathbf{M}_c, \mathbf{F}_\theta(x_j)) - \delta)}}{|\mathcal{D}_{c'}|} - e^{-\beta(d(\mathbf{M}_c, \mathbf{M}_{c'}) - \delta)} \right].
\end{aligned}
\tag{17}
$$

Note that the negative term in Eq. (10) can be ignored due to the mean field regularization.

## A.4 APPLICATION TO PAIR-BASED LOSSES WITH ANCHORS AND COMPARISION WITH THE PROXY-BASED METHOD

In the main part of this paper, we focused on the application of mean field theory to losses without anchors. We here apply it to those with anchors, which typically take the following form:

$$
\begin{aligned}
\mathcal{L} = & \overline{\sum_{i \in \mathcal{D}}} \, \overline{\sum_{P_i \subset \mathcal{D}}} \, \overline{\sum_{N_i \subset \mathcal{D}}} l(\mathbf{F}_\theta(x_i), \{\mathbf{F}_\theta(x_j)\}_{j \in P_i}, \{\mathbf{F}_\theta(x_k)\}_{k \in N_i}) \\
= & \overline{\sum_{i \in \mathcal{D}}} \, \overline{\sum_{P_i \subset \mathcal{D}}} \, \overline{\sum_{N_i \subset \mathcal{D}}} l(\{d(\mathbf{F}_\theta(x_i), \mathbf{F}_\theta(x_j)\}_{j \in P_i}, \{d(\mathbf{F}_\theta(x_i), \mathbf{F}_\theta(x_k)\}_{k \in N_i}),
\end{aligned}
\tag{18}
$$

where $x_i$, $P_i$, and $N_i$ represent an anchor, a set of positive samples, and a set of negative samples, respectively. Note that we also introduced a normalized sum, $\overline{\sum_a} f(a) = \sum_a f(a) / \sum_a 1$ for simplicity. We then expand this loss function with respect to $(\mathbf{M}_c - \mathbf{F}_\theta(x_i))$. Since Eq. (18) depends only on the distances between anchors and positive or negative samples, we must ignore cross terms of fluctuations of anchors and those of positive and negative samples. Taking the resummation over the remaining terms, we obtain

$$
\mathcal{L} \simeq \mathcal{L}_{\mathrm{MFT}} = \frac{1}{|\mathcal{C}|} \sum_{c \in \mathcal{C}} \frac{1}{|\mathcal{D}_c|} \sum_{i \in \mathcal{D}_c} l(\mathbf{F}_\theta(x_i), M_c, \{M_{c'}\}_{c'(\neq c)}) + \overline{\sum_{c \in \mathcal{C}}} \, \overline{\sum_{P \subset \mathcal{D}_c}} \, \overline{\sum_{N \subset \mathcal{D} \backslash \mathcal{D}_c}} l(M_c, \{\mathbf{F}_\theta(x_j)\}_{j \in P_i}, \{\mathbf{F}_\theta(x_k)\}_{k \in N_i}),
\tag{19}
$$

**Algorithm 1** Mean field losses training procedure.

1: Initialize parameters $\theta$ and mean fields $\mathbf{M}$ randomly.
2: **for** step $= 1, \cdots, N_{\text{steps}}$ **do**
3:     Sample a batch $\mathcal{B} = \{x_i, y_i\}_i$ from dataset $\mathcal{D}$
4:     Compute $\mathcal{L}_{\text{MF}}$ for batch $\mathcal{B}$ and classes $\mathcal{C}$
5:     Update $\theta$ and $\mathbf{M}$ using gradients of $\mathcal{L}_{\text{MF}}$
6: **end for**

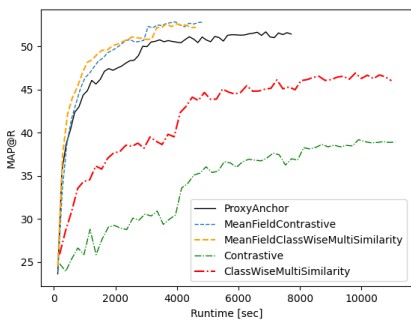

Figure 4: Accuracy (MAP@R) vs. runtime comparison on the SOP dataset.

with the constraint $\{l(M_c, M_c, \{M_{c'}\}_{c'(\neq c)}) = 0\}_{c \in \mathcal{C}}$. While the proxy-based approach (Movshovitz-Attias et al., 2017) yields only the first term, our approach also gives us the second term, which represents interactions between a mean field and multiple positive and negative samples. This term is analogous to ProxyAnchor loss (Kim et al., 2020) and introduces interactions between samples. Furthermore, the second term comes from the expansion series with respect to the fluctuations of positive and negative data points, which are ignored in the proxy-based losses. Therefore, we can expect that our mean field loss approximates the original pair-based loss better than the proxy-based one near the optimal solution.

For instance, let us consider NCA loss (Goldberger et al., 2004), which takes the following form:

$$\mathcal{L}_{\text{NCA}} = -\overline{\sum_{i \in \mathcal{D}}} \overline{\sum_{P_i \subset \mathcal{D}}} \overline{\sum_{N_i \subset \mathcal{D}}} \log \left[ \frac{\sum_{j \in P_i} e^{-\alpha d(\mathbf{F}_\theta(x_i), \mathbf{F}_\theta(x_j))}}{\sum_{j \in P_i} e^{-\alpha d(\mathbf{F}_\theta(x_i), \mathbf{F}_\theta(x_j))} + \sum_{k \in N_i} e^{-\alpha d(\mathbf{F}_\theta(x_i), \mathbf{F}_\theta(x_k))}} \right]. \quad (20)$$

Applying the above Eq. (18) leads to MeanFieldNCA loss

$$\begin{aligned}
\mathcal{L}_{\text{NCA}} \simeq \mathcal{L}_{\text{MFNCA}} = &-\frac{1}{|\mathcal{C}|} \sum_{c \in \mathcal{C}} \frac{1}{|\mathcal{D}_c|} \sum_{i \in \mathcal{D}_c} \log \left[ \frac{e^{-\alpha d(\mathbf{F}_\theta(x_i), \mathbf{M}_c)}}{\sum_{c' \in \mathcal{C}} e^{-\alpha d(\mathbf{F}_\theta(x_i), \mathbf{M}_{c'})}} \right] \\
&- \overline{\sum_{c \in \mathcal{C}}} \overline{\sum_{P \subset \mathcal{D}_c}} \overline{\sum_{N \subset \mathcal{D} \setminus \mathcal{D}_c}} \log \left[ \frac{\sum_{j \in \mathcal{D}_c} e^{-\alpha d(\mathbf{M}_c, \mathbf{F}_\theta(x_i))}}{\sum_{j \in P} e^{-\alpha d(\mathbf{M}_c, \mathbf{F}_\theta(x_j))} + \sum_{k \in N} e^{-\alpha d(\mathbf{M}_c, \mathbf{F}_\theta(x_k))}} \right].
\end{aligned} \quad (21)$$

Here, the first term is equal to ProxyNCA loss and describes an interaction with the mean fields and each sample. On the other hand, the second term incorporates interactions among samples like ProxyAnchor and MFCWMS losses.

## A.5   Pseudocode and runtime

Algorithm 1 delineates the procedure for updating the embedding model $F_\theta(x)$ and mean fields $\mathbf{M}_c$ during training. Focusing on the $T = 0$ scenario as discussed in Section 3.1, we employ a similar optimization strategy for our learnable parameters as utilized in other classification-based loss functions. This approach results in enhanced convergence efficiency, measured in runtime as well, as demonstrated in Figure 4.

Lastly, let us mention how we optimize the partition function $-\log Z_{\text{MFT}}$ for $T > 0$ in deep metric learning. In this case, the partition function can be written as

$$-\log Z_{\text{MFT}}(\mathbf{M}) = -\log \int d\theta e^{-\mathcal{L}_{\text{MFT}}(\theta, \mathbf{M}, \mathcal{D})/T}, \quad (22)$$

which is similar to the objective function in the expectation–maximization (EM) algorithm. To facilitate optimization, we introduce a distribution function $Q(\theta)$ and minimize the following expression instead of $-\log Z_{\mathrm{MFT}}$:

$$
\begin{aligned}
\mathcal{F}(Q, \mathbf{M}) &= -\int d\theta Q(\theta) \log \frac{e^{-\mathcal{L}_{\mathrm{MFT}}(\theta, \mathbf{M}, \mathcal{D})/T}}{Q(\theta)} \\
&= -\int d\theta Q(\theta) \log \frac{e^{-\mathcal{L}_{\mathrm{MFT}}(\theta, \mathbf{M}, \mathcal{D})/T}/Z_{\mathrm{MFT}}(\mathbf{M})}{Q(\theta)} - \log Z_{\mathrm{MFT}}(\mathbf{M}) \\
&= -\log Z_{\mathrm{MFT}}(\mathbf{M}) + \mathrm{KL}\left(Q(\theta) || \frac{e^{-\mathcal{L}_{\mathrm{MFT}}(\theta, \mathbf{M}, \mathcal{D})/T}}{Z_{\mathrm{MFT}}(\mathbf{M})}\right).
\end{aligned}
\tag{23}
$$

Thanks to $\mathrm{KL}(Q||P) \geq 0$, the lower bound for its minimum is the partition function itself. Minimizing $\mathcal{F}(Q, \mathbf{M})$ with respect to $Q(\theta)$ (the E step) yields $Q(\theta) = e^{-\mathcal{L}_{\mathrm{MFT}}(\theta, \mathbf{M}, \mathcal{D})/T}/Z_{\mathrm{MFT}}(\mathbf{M})$. Conversely, minimizing $\mathcal{F}(Q, \mathbf{M})$ with respect to $\mathbf{M}$ (the M step) leads to the consistency condition $\mathbf{M}_c = \mathbb{E}_Q[\sum_{i \in \mathcal{D}_c} \mathbf{F}_\theta(x_i)/|\mathcal{D}_c|]$ at the first order of the expansion. Iteratively performing the E and M steps until convergence results in optimal $\mathbf{M}$ and $Q(\theta)$.

# B  ADDITIONAL EXPERIMENTAL RESULTS

In this section, we present the experimental results that cannot be shown in the main paper due to the page limit.

## B.1  MLRC RESULTS

Tables 3 – 5 show the complete results of Table 1 in the main paper, which are obtained the modern benchmark protocol proposed in the "Metric Learning Reality Check" (MLRC) paper (Musgrave et al., 2020a). In the CUB-200-2011 (CUB) dataset (Wah et al., 2011), MeanFieldContrastive (MFCont.) and MeanFieldClassWiseMultiSimilarity (MFCWMS) losses outperform the others in Mean Average Precision at R (MAP@R) and R-Precision (RP), while ProxyAnchor loss (Kim et al., 2020) is better in Precision at 1 (P@1) in the separated case. In contrast, in the Stanford Online Products (SOP) dataset (Oh Song et al., 2016), the MFCWMS loss shows the best performance in all the metrics.

Table 3: MLRC evaluation results in CUB. We carry out 10 test runs and show averaged metrics with their confidence intervals.

| Loss | Separated (128D) | | | Concatenated (512D) | | |
|------|---------|---------|---------|---------|---------|---------|
| | MAP@R | P@1 | RP | MAP@R | P@1 | RP |
| ArcFace | $21.46 \pm 0.13$ | $59.98 \pm 0.22$ | $32.31 \pm 0.14$ | $26.39 \pm 0.16$ | $67.11 \pm 0.23$ | $37.23 \pm 0.17$ |
| CosFace | $21.19 \pm 0.22$ | $59.74 \pm 0.28$ | $32.00 \pm 0.23$ | $\underline{26.54 \pm 0.29}$ | $67.14 \pm 0.29$ | $\underline{37.38 \pm 0.28}$ |
| MS | $20.98 \pm 0.16$ | $59.38 \pm 0.27$ | $31.84 \pm 0.15$ | $26.20 \pm 0.16$ | $67.34 \pm 0.35$ | $36.99 \pm 0.16$ |
| MS+Miner | $20.78 \pm 0.17$ | $59.02 \pm 0.25$ | $31.67 \pm 0.16$ | $25.94 \pm 0.18$ | $67.08 \pm 0.32$ | $36.77 \pm 0.16$ |
| ProxyNCA | $18.75 \pm 0.18$ | $57.06 \pm 0.27$ | $29.64 \pm 0.21$ | $23.84 \pm 0.22$ | $65.60 \pm 0.28$ | $34.82 \pm 0.25$ |
| ProxyAnch. | $\underline{21.67 \pm 0.22}$ | $\mathbf{60.80 \pm 0.33}$ | $\underline{32.53 \pm 0.23}$ | $26.48 \pm 0.23$ | $\underline{67.72 \pm 0.30}$ | $37.30 \pm 0.23$ |
| Cont. | $21.02 \pm 0.14$ | $59.35 \pm 0.33$ | $31.80 \pm 0.15$ | $26.37 \pm 0.18$ | $\underline{67.67 \pm 0.25}$ | $37.10 \pm 0.19$ |
| MFCont. | $\underline{22.01 \pm 0.10}$ | $\underline{60.29 \pm 0.23}$ | $\underline{32.85 \pm 0.10}$ | $\mathbf{27.16 \pm 0.07}$ | $67.64 \pm 0.27$ | $\mathbf{37.95 \pm 0.07}$ |
| CWMS | $21.48 \pm 0.27$ | $60.09 \pm 0.27$ | $32.32 \pm 0.26$ | $26.94 \pm 0.29$ | $\underline{68.24 \pm 0.42}$ | $37.69 \pm 0.27$ |
| MFCWMS | $\mathbf{22.11 \pm 0.08}$ | $\underline{60.28 \pm 0.10}$ | $\mathbf{32.96 \pm 0.08}$ | $\underline{27.03 \pm 0.12}$ | $67.63 \pm 0.21$ | $\underline{37.83 \pm 0.12}$ |

Table 4: MLRC evaluation results in Cars. We carry out 10 test runs and show averaged metrics with their confidence intervals.

| Loss | Separated (128D) | | | Concatenated (512D) | | |
|------|---------|---------|---------|---------|---------|---------|
| | MAP@R | P@1 | RP | MAP@R | P@1 | RP |
| ArcFace | $18.25 \pm 0.12$ | $71.12 \pm 0.36$ | $28.63 \pm 0.13$ | $\underline{27.63 \pm 0.15}$ | $84.39 \pm 0.15$ | $\underline{37.45 \pm 0.15}$ |
| CosFace | $18.49 \pm 0.13$ | $74.66 \pm 0.21$ | $28.75 \pm 0.12$ | $26.96 \pm 0.25$ | $85.29 \pm 0.26$ | $36.80 \pm 0.24$ |
| MS | $18.66 \pm 0.30$ | $71.89 \pm 0.33$ | $29.42 \pm 0.29$ | $27.19 \pm 0.41$ | $84.03 \pm 0.30$ | $37.39 \pm 0.36$ |
| MS+Miner | $18.49 \pm 0.23$ | $71.99 \pm 0.28$ | $29.20 \pm 0.23$ | $26.89 \pm 0.38$ | $83.89 \pm 0.36$ | $37.09 \pm 0.33$ |
| ProxyNCA | $17.43 \pm 0.11$ | $70.96 \pm 0.26$ | $27.85 \pm 0.10$ | $26.78 \pm 0.18$ | $84.31 \pm 0.24$ | $36.83 \pm 0.17$ |
| ProxyAnch. | $\mathbf{19.44 \pm 0.17}$ | $\mathbf{76.15 \pm 0.25}$ | $\underline{29.89 \pm 0.18}$ | $26.81 \pm 0.27$ | $\mathbf{85.53 \pm 0.30}$ | $36.76 \pm 0.26$ |
| Cont. | $17.04 \pm 0.26$ | $69.77 \pm 0.40$ | $27.48 \pm 0.26$ | $24.93 \pm 0.46$ | $81.87 \pm 0.35$ | $35.12 \pm 0.42$ |
| MFCont. | $\underline{18.12 \pm 0.13}$ | $\underline{71.77 \pm 0.28}$ | $\underline{28.54 \pm 0.14}$ | $\underline{27.37 \pm 0.18}$ | $\underline{84.56 \pm 0.21}$ | $\underline{37.19 \pm 0.18}$ |
| CWMS | $\underline{19.27 \pm 0.26}$ | $\underline{74.19 \pm 0.30}$ | $\mathbf{29.95 \pm 0.25}$ | $\mathbf{27.80 \pm 0.33}$ | $85.18 \pm 0.28$ | $\mathbf{37.89 \pm 0.29}$ |
| MFCWMS | $18.85 \pm 0.16$ | $73.02 \pm 0.20$ | $29.55 \pm 0.15$ | $26.98 \pm 0.31$ | $84.00 \pm 0.22$ | $37.11 \pm 0.27$ |

Table 5: MLRC evaluation results in SOP. We carry out 10 test runs and show averaged metrics with their confidence intervals. We removed ProxyAnchor because it failed to converge in our settings.

| Loss | Separated (128D) | | | Concatenated (512D) | | |
|------|---------|---------|---------|---------|---------|---------|
| | MAP@R | P@1 | RP | MAP@R | P@1 | RP |
| ArcFace | $41.47 \pm 0.24$ | $71.39 \pm 0.20$ | $44.35 \pm 0.23$ | $\underline{47.37 \pm 0.23}$ | $\underline{76.13 \pm 0.16}$ | $\underline{50.22 \pm 0.22}$ |
| CosFace | $41.01 \pm 0.24$ | $71.03 \pm 0.22$ | $43.89 \pm 0.24$ | $46.77 \pm 0.20$ | $75.69 \pm 0.13$ | $49.63 \pm 0.20$ |
| MS | $41.87 \pm 0.21$ | $71.10 \pm 0.18$ | $45.00 \pm 0.20$ | $46.70 \pm 0.18$ | $75.21 \pm 0.15$ | $49.70 \pm 0.17$ |
| MS+Miner | $41.90 \pm 0.30$ | $71.08 \pm 0.25$ | $45.05 \pm 0.30$ | $46.57 \pm 0.28$ | $75.09 \pm 0.19$ | $49.57 \pm 0.28$ |
| ProxyNCA | $\underline{42.73 \pm 0.11}$ | $\underline{71.77 \pm 0.08}$ | $\underline{45.72 \pm 0.11}$ | $46.73 \pm 0.13$ | $75.24 \pm 0.10$ | $49.61 \pm 0.13$ |
| Cont. | $41.09 \pm 0.18$ | $70.04 \pm 0.16$ | $44.18 \pm 0.19$ | $45.35 \pm 0.19$ | $73.88 \pm 0.15$ | $48.28 \pm 0.19$ |
| MFCont. | $\underline{43.62 \pm 0.36}$ | $\underline{72.74 \pm 0.29}$ | $\underline{46.55 \pm 0.35}$ | $\underline{47.01 \pm 0.21}$ | $\underline{75.57 \pm 0.16}$ | $\underline{49.85 \pm 0.20}$ |
| CWMS | $41.53 \pm 0.20$ | $70.76 \pm 0.16$ | $44.50 \pm 0.21$ | $45.13 \pm 0.16$ | $73.99 \pm 0.11$ | $47.99 \pm 0.16$ |
| MFCWMS | $\mathbf{44.57 \pm 0.16}$ | $\mathbf{73.32 \pm 0.11}$ | $\underline{47.53 \pm 0.16}$ | $\mathbf{48.33 \pm 0.18}$ | $\mathbf{76.38 \pm 0.14}$ | $\mathbf{51.17 \pm 0.18}$ |

Moreover, we performed the T-test for MAP@R of MFCont. and MFCWMS losses against ArcFace and ProxyAnchor losses at the $95\%$ confidence level as shown in Tables. 6 and 7. MFCWMS loss shows greater performance in all scenarios in CUB and

Table 6: T-test results for MAP@R of the mean field losses compared to ArcFace loss are presented. ✓ indicates statistical significance, while × denotes non-significance. Additionally, ↑ signifies that the mean field losses outperform ArcFace loss, whereas ↓ indicates the opposite.

| Loss | CUB | | Cars | | SOP | |
|---|---|---|---|---|---|---|
| | **128D** | **512D** | **128D** | **512D** | **128D** | **512D** |
| MFCont. | ✓ ↑ | ✓ ↑ | × ↓ | ✓ ↓ | ✓ ↑ | ✓ ↓ |
| MFCWMS | ✓ ↑ | ✓ ↑ | ✓ ↑ | ✓ ↓ | ✓ ↑ | ✓ ↑ |

Table 7: T-test results for MAP@R of the mean field losses compared to ProxyAnchor loss are presented. ✓ indicates statistical significance, while × denotes non-significance. Additionally, ↑ signifies that the mean field losses outperform ProxyAnchor loss, whereas ↓ indicates the opposite.

| Loss | CUB | | Cars | | SOP | |
|---|---|---|---|---|---|---|
| | **128D** | **512D** | **128D** | **512D** | **128D** | **512D** |
| MFCont. | ✓ ↑ | ✓ ↑ | ✓ ↓ | ✓ ↑ | − | − |
| MFCWMS | ✓ ↑ | ✓ ↑ | ✓ ↓ | × ↑ | − | − |

Table 8: Impact of batch size to MAP@R for MFCWMS loss in CUB and Cars.

| Batch size | CUB | Cars |
|---|---|---|
| 30 | $23.3 \pm 0.3$ | $22.0 \pm 0.3$ |
| 60 | $24.4 \pm 0.1$ | $23.2 \pm 0.3$ |
| 90 | $25.1 \pm 0.1$ | $23.6 \pm 0.4$ |
| 120 | $25.3 \pm 0.2$ | $23.9 \pm 0.2$ |
| 150 | $25.4 \pm 0.3$ | $24.2 \pm 0.3$ |
| 180 | $25.4 \pm 0.2$ | $24.2 \pm 0.2$ |

Table 9: Impact of batch size to MAP@R for MFCWMS loss in SOP and InShop.

| Batch size | SOP | InShop |
|---|---|---|
| 30 | $51.1 \pm 0.1$ | $67.4 \pm 0.2$ |
| 60 | $52.0 \pm 0.1$ | $67.6 \pm 0.2$ |
| 90 | $52.4 \pm 0.1$ | $67.6 \pm 0.2$ |
| 120 | $52.7 \pm 0.1$ | $67.8 \pm 0.2$ |
| 150 | $52.8 \pm 0.1$ | $67.0 \pm 0.6$ |
| 300 | $53.0 \pm 0.3$ | $67.0 \pm 0.4$ |

SOP datasets. Additionally, our simpler loss function, MFCont loss, although it is slightly less competitive than MFCWMS loss, still holds up well against these benchmarks.

## B.2 BATCH SIZE

We also investigate how batch size affects the accuracy of our models. We compare the test performance of MFCWMS loss by changing the batch size between 30 and 180 for CUB and Cars (Table 8) and 30 and 300 for SOP and InShop (Liu et al., 2016) (Table 9). The accuracy metric basically improves as the batch size increases. The improvement mostly saturates around 150 and 180 for the smaller datasets and around 150 and 300 for SOP. In the InShop dataset, we observe that the accuracy drops around batch size 150 with a relatively large variance. Note that the accuracy of ProxyAnchor loss in the InShop dataset also saturates between 60 and 150 and starts to decrease gradually for large batch sizes. Moreover, Table 11 shows the MAP@R in ProxyAnchor and MFCWMS for the InShop dataset without the query–gallery split of test data. Accuracies of both losses start to decrease gradually around batch size 150, which is consistent with Table 10.

## B.3 $\alpha$ OF MFCWMS

In addition to $\beta$ and $\delta$ of MFCWMS loss, we studied the effect of $\alpha$. We varied $\alpha$ from 0.01 to 40, fixing $(\beta, \delta, \lambda_{\mathrm{MF}})$ to $(80, 0.8, 0)$ and computed the MAP@R and the best epoch in the CUB dataset. The result is summarized in Table. 12 and indicates the performance of MFCWMS loss glows slowly as $\alpha$ increases until $\alpha \simeq 20$ while the convergence speed becomes slower for larger $\alpha$.

Table 10: Accuracies in InShop with the query–gallery split.

| Batch size | ProxyAnchor | MFCWMS |
|---|---|---|
| 30 | $63.6 \pm 1.4$ | $67.4 \pm 0.2$ |
| 60 | $\underline{65.7 \pm 0.2}$ | $67.6 \pm 0.2$ |
| 90 | $65.5 \pm 0.3$ | $67.6 \pm 0.2$ |
| 120 | $65.6 \pm 0.3$ | $\underline{67.8 \pm 0.2}$ |
| 150 | $65.5 \pm 0.2$ | $67.0 \pm 0.6$ |
| 300 | $64.5 \pm 0.2$ | $67.0 \pm 0.4$ |
| 500 | $63.3 \pm 0.2$ | $67.1 \pm 0.1$ |

Table 11: Accuracies in InShop *without* the query–gallery split.

| Batch size | ProxyAnchor | MFCWMS |
|---|---|---|
| 30 | $61.7 \pm 0.6$ | $64.7 \pm 0.1$ |
| 60 | $62.8 \pm 0.4$ | $\underline{65.1 \pm 0.2}$ |
| 90 | $62.9 \pm 0.3$ | $65.0 \pm 0.5$ |
| 120 | $\underline{62.9 \pm 0.3}$ | $64.9 \pm 0.5$ |
| 150 | $62.7 \pm 0.1$ | $65.0 \pm 0.4$ |
| 300 | $61.9 \pm 0.2$ | $64.7 \pm 0.4$ |
| 500 | $60.6 \pm 0.2$ | $64.6 \pm 0.1$ |

Table 12: MAP@R values and epochs with the best accuracies against the hyperparamter $\alpha$ of MFCWMS loss in the CUB dataset.

| $\alpha$ | 0.01 | 0.1 | 1 | 10 | 20 | 40 |
|---|---|---|---|---|---|---|
| MAP@R | $25.3 \pm 0.3$ | $25.3 \pm 0.3$ | $25.4 \pm 0.3$ | $\underline{25.5 \pm 0.3}$ | $\underline{25.5 \pm 0.2}$ | $24.8 \pm 0.2$ |
| Epoch | $5.0 \pm 0.9$ | $4.8 \pm 1.1$ | $\underline{4.7 \pm 0.8}$ | $8.7 \pm 1.9$ | $14.0 \pm 2.1$ | $16.7 \pm 4.5$ |

Table 13: MAP@R against the regularization coefficient $\lambda_{\mathrm{MF}}$ in Eq. (10) in the CUB dataset.

| $\lambda_{\mathrm{MF}}$ | 0 | 0.01 | 0.1 | 1 | 10 | 100 |
|---|---|---|---|---|---|---|
| MFCont. | $25.2 \pm 0.3$ | $25.2 \pm 0.3$ | $\underline{25.3 \pm 0.3}$ | $25.2 \pm 0.2$ | $25.2 \pm 0.3$ | $25.0 \pm 0.3$ |
| MFCWMS | $\underline{25.3 \pm 0.2}$ | $\underline{25.3 \pm 0.3}$ | $\underline{25.3 \pm 0.3}$ | $\underline{25.3 \pm 0.3}$ | $25.2 \pm 0.3$ | $25.1 \pm 0.3$ |

### B.4 MEAN FIELD REGULARIZATION

We also studied the impact of the regularization term in Eq. (10). We investigated the test performance of MFCont. and MFCWMS losses in the CUB dataset varying the scale of the regularization coefficient and summarized in Table 13. The results indicate that the regularization does not offer a statistically significant difference in performance. This is because the constraint (7) should be satisfied naturally when the main parts of the losses are minimized. This point can be confirmed by Figs. 5a and 5b, where we plot the values of the mean field regularization terms of MFCont. and MFCWMS losses without $\lambda_{\mathrm{MF}}$ during the training in the CUB dataset. These figures show that the values become zero after two epochs even for $\lambda_{\mathrm{MF}} = 0$. Therefore, we conclude that the constraints in MFCont. and MFCWMS losses are significant theoretically, but less relevant in practice.

### B.5 LEARNING CURVES

Figure 6 shows learning curves obtained in the traditional evaluation protocol (Movshovitz-Attias et al., 2017; Kim et al., 2020) in fixed seeds. Both MFCont. and MFCWMS losses show faster convergence than ProxyAnchor loss as well as Contrastive and CWMS losses. In the smaller datasets (CUB and Cars), accuracies of our mean field losses seem to decrease faster while we don't see such behaviors in the larger datasets (SOP and InShop). This phenomenon might be caused by strong repulsive interactions with negative mean fields. For larger datasets, the embedding spaces may be sufficiently populated to balance the repulsive force, while this may not be the case for smaller datasets. It might not occur for ProxyAnchor loss since repulsive forces for ProxyAnchor loss are weighted depending on distances between proxy and negative samples.

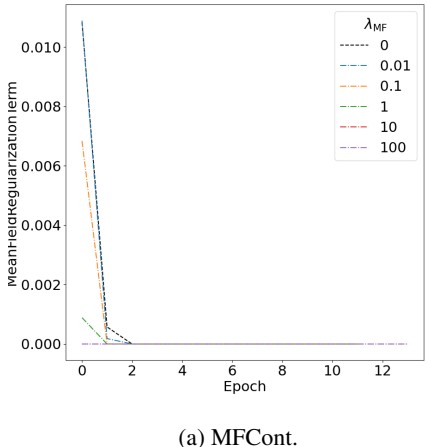

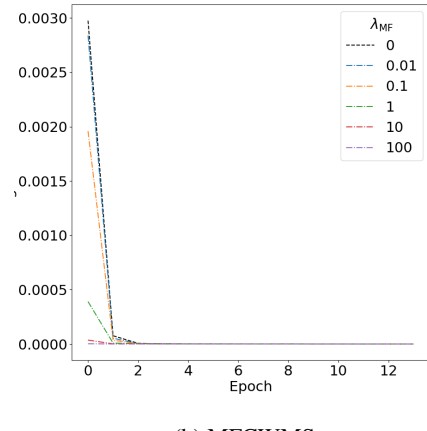

(a) MFCont.                          (b) MFCWMS

Figure 5: Plot of the Mean Field regularization term of (a) MFCont. and (b) MFCWMS ($\lambda_{\mathrm{MF}}$ is not included) in CUB-200-2011. Even if $\lambda_{\mathrm{MF}} = 0$, the constraint imposed by the Mean Field regularization is satisfied in a few epochs.

Table 14: MAP@R values and epochs with the best accuracies obtained using the traditional protocol with VisionTransformer (ViTB16) in the CUB, Cars, SOP, and InShop datasets. The best result in each column is underlined.

| Loss | CUB | | Cars | | SOP | | InShop | |
|---|---|---|---|---|---|---|---|---|
| | MAP@R | Epoch | MAP@R | Epoch | MAP@R | Epoch | MAP@R | Epoch |
| ProxyAnch. | $30.1 \pm 0.5$ | $3.1 \pm 0.7$ | $27.2 \pm 0.2$ | $6.6 \pm 0.8$ | $56.8 \pm 0.3$ | $25.3 \pm 3.6$ | $66.4 \pm 0.1$ | $31.9 \pm 5.9$ |
| MFCont. | $29.2 \pm 0.3$ | $\underline{2.4 \pm 0.4}$ | $26.9 \pm 0.1$ | $\underline{5.6 \pm 0.4}$ | $\underline{58.2 \pm 0.1}$ | $\underline{8.2 \pm 0.5}$ | $68.1 \pm 0.2$ | $\underline{11.9 \pm 1.1}$ |
| MFCWMS | $\underline{30.3 \pm 0.3}$ | $5.1 \pm 0.6$ | $\underline{27.4 \pm 0.2}$ | $8.7 \pm 1.4$ | $57.9 \pm 0.1$ | $12.2 \pm 2.2$ | $\underline{69.4 \pm 0.1}$ | $26.2 \pm 2.3$ |

## B.6 TRADITIONAL EVALUATION WITH VISIONTRANSFORMER

We also performed the traditional evaluation protocol replacing the BN-Inception (Ioffe & Szegedy, 2015) by VisionTransformer (ViTB16) (Dosovitskiy et al., 2020) pretrained on ImageNet (Russakovsky et al., 2015) in the four datasets described in Sec. 4.1. As shown in Table 14, MFCWMS loss outperforms ProxyAnchor loss in MAP@R in all the datasets, and the improvement in accuracy is evident in the larger datasets as observed in Sec. 4.3. On the other hand, MFCWMS loss converges relatively slower in the CUB and Cars datasets. Note that we set $\alpha = 20$ for MFCWMS loss in these experiments.

## B.7 PROXYNCA VS. MEANFIELDNCA

Lastly, we also evaluated ProxyNCA and MFNCA losses in the CUB dataset in the modern benchmark protocol to compare losses from the proxy-based and mean field theory approaches. Table 15 summarises the results and shows MFNCA loss outperforms ProxyNCA loss consistently. In particular, the P@1 is improved better compared to the other metrics. The experimental result implies the better approximation of original loss functions discussed in Sec. A.4 is likely a key to enhancing overall performance in deep metric learning.

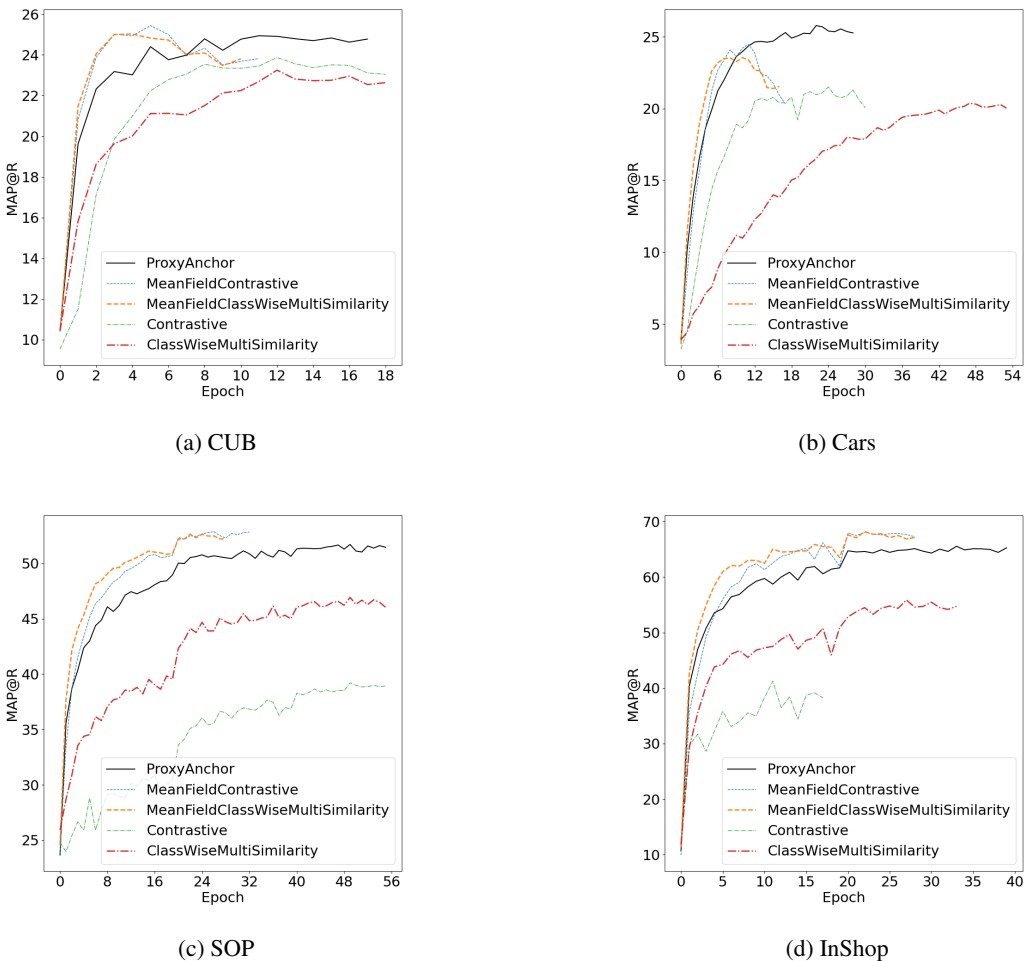

Figure 6: The test accuracy (MAP@R) plotted against the number of epochs for the (a) CUB, (b) Cars, (c) SOP, and (d) InShop datasets, comparing ProxyAnchor, MFCont., MFCWMS, Cont., and CWMS.

Table 15: Comparison of MLRC evaluation results of ProxyNCA and MFNCA losses in the CUB dataset. We carry out 10 test runs and show averaged metrics with their confidence intervals.

| Loss | 128D | | | 512D | | |
|---|---|---|---|---|---|---|
| | MAP@R | P@1 | RP | MAP@R | P@1 | RP |
| ProxyNCA | $18.8 \pm 0.2$ | $57.1 \pm 0.3$ | $29.6 \pm 0.2$ | $23.8 \pm 0.2$ | $65.6 \pm 0.3$ | $34.8 \pm 0.2$ |
| MFNCA | $19.0 \pm 0.2$ | $57.8 \pm 0.3$ | $29.9 \pm 0.2$ | $24.2 \pm 0.3$ | $66.2 \pm 0.4$ | $35.1 \pm 0.3$ |