# OpenReview forum: "Mean Field Theory in Deep Metric Learning"
_ICLR.cc/2024/Conference — ICLR 2024 poster_

### Official Review · Reviewer_q1qw · 2023-10-26

**Soundness:** 2 fair
**Presentation:** 2 fair
**Contribution:** 2 fair
**Rating:** 5
**Confidence:** 5

**Summary:**

In this paper, the mean field theory is introduced into the domain of deep metric learning. By incorporating foundational components such as the Contrastive loss and Class Wise Multi-Similarity loss, the authors construct the Mean Field Contrastive loss and Mean Field Class Wise Multi-Similarity loss. The proposed method is evaluated through extensive experiments on benchmark datasets, covering two benchmark protocols. The results demonstrate the effectiveness of the proposed approach.

**Strengths:**

1. The authors have integrated the mean field theory into the realm of deep metric learning.

2. They have introduced two pair-based loss functions, namely the Mean Field Contrastive loss and the Mean Field Class Wise Multi-Similarity loss.

3. The experiments and ablation studies conducted in the paper are comprehensive.

**Weaknesses:**

Deep metric learning involves a range of loss functions, and it is unclear whether the mean field theory can be applied to other loss functions commonly used in this context. It would be valuable for the authors to specify under what conditions and contexts the mean field theory is applicable and offer practical guidance for its implementation in other scenarios.

**Questions:**

How can the mean field theory be extended to encompass other commonly used loss functions in deep metric learning?

---

> ### Author Response · Authors · 2023-11-16
> **Thank you for your review!**
>
> Thank you for your insightful comments and the opportunity to discuss the extensions of our work. We are grateful for the chance to elaborate on how Mean-Field Theory (MFT) can further enhance deep metric learning.
>
> To address your question about extending MFT to other common loss functions in deep metric learning, we have added theoretical insights into applying MFT to losses with anchors in Section A.4 in the appendix.
>
> Unlike the proxy-based method, which replaces positive and negative samples with proxies, the mean field approach results in two distinct terms. In one term, positive and negative samples are replaced by mean fields, similar to the proxy-based approach. However, in the other term, anchors are replaced by mean fields while the positive and negative samples remain. This term is analogous to the ProxyAnchor loss, which was introduced heuristically.
>
> We hope this response adequately addresses your query and provides further clarity on our work. We are thankful for your engagement with our research and welcome any additional comments or questions you may have.

---

> > ### Author Response · Authors · 2023-11-21
> > **On updates in our revised manuscript**
> >
> > We have carefully considered your comments and have made the following significant updates to address your concerns:
> >
> > **Application to NCA Loss**
> >
> > In response to your query on applying Mean-Field Theory (MFT) to other common loss functions in deep metric learning, we have demonstrated the application of MFT to the NCA loss and derived Mean Field NCA (MFNCA) loss (Eq. (21) of our revised manuscript). We think this extension shows the versatility of MFT in the context of deep metric learning and provides a practical guidance for its implementation in various scenarios.
> >
> > **Benchmarking MFNCA loss**
> >
> > To empirically validate the effectiveness of the MFNCA loss, we conducted an MLRC benchmark on the CUB dataset. The results, as detailed in Table 15 of Section B.7, show a consistent improvement in accuracy metrics. This experiment suggests that the additional term derived from the expansions with respect to positive and negative samples contributes to a more accurate approximation of the original loss functions, indicating the utility of MFT in losses with anchors.
> >
> > We believe these updates comprehensively address your concerns and further strengthen our manuscript. We remain open to any additional suggestions or queries you might have and look forward to your feedback.

---

> > > ### Comment · Reviewer_q1qw · 2023-11-22
> > > **Thank you for the respense**
> > >
> > > I lean more towards the comments of Reviewer zMTp and Reviewer aqYA. Currently, I will maintain my rating.

---

### Official Review · Reviewer_pubU · 2023-10-31

**Soundness:** 3 good
**Presentation:** 3 good
**Contribution:** 3 good
**Rating:** 6
**Confidence:** 3

**Summary:**

The paper considers mean field theory approximations of pair-based loss functions for metric learning. Inspired by techniques in statistical physics, pairwise calculations are approximated by comparing to a mean approximation, ie, turning a summation over $i$ and $j$ to only that of $i$. Using such an approach, two different mean field contrastive loss functions are proposed. Empirically, the proposed loss functions are evaluated and are shown to even out perform their non-mean field counterparts.

**Strengths:**

- The approximation technique for reducing pairwise summations to their mean field approximation intuitively makes sense. This concept is particularly well illustrated in Figure 1.
- The proposed approach seems promising. Surprisingly, the mean field approximations perform better than their non-mean field counterparts in many circumstances.

**Weaknesses:**

- Part of the motivation for the approximation (not including its statistical physics analogy) was the reduction in runtime complexity pair-based loss function in metric learning. However, there is no runtime values reported in the paper.
- Some terms in the paper are not explained. (See questions below).

**Questions:**

- What are the runtimes of the mean field variants compared to their regular counterparts? Does the additional complexity of requiring optimization of mean fields $\mathbf{M}_c$ outweigh the reduction of computational complexity via the mean field approximation?
- One of the major interesting components of the paper is that mean field approximation performs better than its non-mean field counterparts. Is there a good hypothesis for why this may be the case? Have you found a good characterization of when one out performs the other? Additional insight here would be great since the paper claims in the empirical section that the mean field losses provide better "training complexity but also results in better embeddings".
- I am unsure exactly what the authors mean be "resummation" and "unstable terms". Clarification here would be great.
- Furthermore, what do that authors mean by "... the above discussion implies the mean field theory is independent of the concept of anchors ..."?

---

> ### Author Response · Authors · 2023-11-16
> **Thank you for your review!**
>
> Thank you for your detailed and constructive feedback on our manuscript. We appreciate the time you have taken to review our work and your insightful comments, which have helped us improve the quality and clarity of our research. Below, we address each of your questions:
>
> **Runtime comparison**
>
> Thank you for highlighting the importance of runtime comparison. We have now included a new figure in our revised manuscript (Fig. 4 in Appendix), which compares the runtimes of the mean field losses against their regular counterparts. As indicated in the figure, the mean field variants not only offer a reduction in computational complexity due to the mean field approximation but also demonstrate faster runtime performance. This improvement aligns with our theoretical expectations and supports the practical applicability of our approach.
>
> **Performance of mean field approximations**
>
> We appreciate your interest in the superior performance of mean field approximations. We hypothesize that the mean field approximation reduces the noise introduced in pairwise comparisons by considering the class means, thus simplifying the optimization process. This could lead to more robust and efficient training, as reflected in our empirical results. As you suggested, we added a sentence
> `This is perhaps because the mean field losses can reduce the noise introduced in pairwise comparisons by the mean fields.` in Sec. 4.3.
>
> **Clarification on 'resummation' and 'unstable terms'**
>
> We realize that these terms were not sufficiently clarified in our original manuscript. Resummation refers to reconverting an infinite series back into a functional form using Taylor expansion inversely. Regarding unstable terms, these mean potential components that could violate the positivity of a loss function. We have addressed this by adding mean field constraints that effectively eliminate such terms. For clarity, additional explanations have been added to the manuscript (footnote 2 on page 5).
>
> **Mean field theory and concept of anchors**
>
> Your question brings to light a crucial aspect of our work. Our approach differs fundamentally from proxy-based methods, which assume a loss depends on the triad of anchor, positive, and negative elements. In contrast, our mean field theory (MFT) approach can be applied to any loss function through Taylor expansion and does not assume a special form of the loss function. This distinct methodology allows for broader applicability and simplicity.
> To clarify this point, we modified the sentence as follows:
> `Furthermore, in contrast to the proxy-based method, which can be applied only to a pair-based loss with an anchor, the mean field theory is applicable to wider types of pair-based loss functions as it is based on the Taylor expansions.`
>
>
> In summary, we hope that our revisions and additional explanations adequately address your questions and enhance the clarity and impact of our work. We are grateful for the opportunity to improve our manuscript based on your feedback and look forward to any further suggestions you may have.

---

> > ### Comment · Reviewer_pubU · 2023-11-21
> >
> > Thank you for your response. The clarification regarding runtime, resummation, and unstable terms was helpful.
> >
> > Regarding the addition explanation for the performance, it seems that an increase robustness is plausible. However, I am hoping that the authors might provide an extended discussion beyond a sentence. I think that even with this sentence, it is hard to justify the MFT approximation except for the computation / runtime improvements, similar to Reviewer zMTp's follow up response.

---

> > > ### Author Response · Authors · 2023-11-21
> > > **Thank you for your reply.**
> > >
> > > Thank you for your continued engagement with our work and for your constructive feedback.
> > > We are happy to hear our modification and clarification were helpful to you.
> > > We also acknowledge your request on the justification of MFT.
> > >
> > > **Additional terms in MFT and their impact**
> > >
> > > As detailed in Section A.4 of the latest manuscript version, Applying MFT to anchor loss introduces terms that are not derived through the proxy-based method. These terms emerge from the expansion with respect to positive/negative samples and are typically neglected in proxy-based approaches.
> > > By incorporating these terms, MFT yields a closer approximation to the original loss function, particularly near the optimal solution. This finer approximation is likely a significant factor contributing to the enhanced performance observed in our experiments, as it provides a more accurate representation of the loss landscape.
> > >
> > > To empirically validate this point, we conducted a benchmark test using the MLRC protocol for our MFNCA loss (the mean field counterpart of ProxyNCA, derived in Eq. (21)) on the CUB dataset.
> > > The results, which we discuss in Section B.7 of our manuscript, demonstrate consistent improvements in accuracy metrics MFNCA compared to ProxyNCA. This empirical evidence further supports our hypothesis that the superior approximation capabilities of MFT lead to improved performance.
> > >
> > > Once again, we are grateful for your insightful feedback and for the opportunity to further refine our manuscript. We look forward to any additional comments or suggestions you may have.

---

### Official Review · Reviewer_uk7M · 2023-10-31

**Soundness:** 4 excellent
**Presentation:** 4 excellent
**Contribution:** 3 good
**Rating:** 8
**Confidence:** 3

**Summary:**

The manuscript proposes two new metric learning algorithms inspired by mean-field analysis from physics. In pair-based algorithms, the loss function pushes pairs of the same class to be closer and pairs of different classes to be apart, and need to be calculated over many training pairs; in contrast, for the proposed mean-field approach, the loss pushes each sample to be close to the class mean and away from other classes' mean, and further pushing class means away from each other. The underlying technical derivation is quite general, through taking a derivating of an energy function. Thus, the same method can be applied to other cases, such as proposing a loss function using class means in a minibatch setting.

**Strengths:**

* Regardless of any inspiration taken from physics, replacing pair-based methods with mean-based methods seems a scalable approach, well grounded in statistics.
 * The proposed class of methods is possibly prudent in the sense that they can be used to derive loss functions, taking into account mean-class information for other problems.
 * The simulations performed seem great, and the authors explain the optimisation carried out on both their method and other methods used for comparison. The results suggest the proposed class of methods achieve very competitive results.
 * The writing is clear, almost tutorial-like, and easy to follow.

**Weaknesses:**

* The authors hide the actual derivation in the appendix, so they do not detail enough their technical approach. On the face of it, the modified loss function could have been suggested just through statistical intuition, not derived from an energy approximation (Hubbard-Stratonovich, saddle-point approx, etc.), so it is a pity the authors don't sketch their methodology in the main text.
 * It is hinted that the method can be more efficient (e.g. due to the lack of pair sampling or anchor points choice), but it is not reported if the training time is superior to the other methods or if the optimisation over `M`-s causes a substantial overhead (which may be justifiable with the improved results).

**Questions:**

* Is it always favourable to optimize the means `M` and the parameters `theta` together? Maybe alternating between optimisation steps would be superior in numerical terms.
 * The method seems similar to classic soft-clustering approaches, which were optimised by alternating between two optimisation steps, such as the EM algorithm. Can you make the connection more explicit?
 * Are the new methods faster or slower than other methods?

---

> ### Author Response · Authors · 2023-11-16
> **Thank you for your review!**
>
> Thank you for your insightful feedback on our manuscript. We appreciate the detailed review and constructive comments. Here are our responses to your points:
>
> **Technical derivation in the main text**
>
> We understand your concern regarding the detailed technical derivation. However, due to page limitations, it is challenging to include the full derivation in the main text without sacrificing other critical content. We have, therefore, made an effort to enhance the clarity and accessibility of the appendix, ensuring that readers can easily follow the derivation process.
>
> **Optimization strategy**
>
> We agree that the optimization of M and $\theta$ is a crucial aspect of our approach.
> For T=0, simultaneous optimization is feasible and effective since the standard stochastic gradient descent is applicable.
> However, for T>0, the optimization becomes more complex, and an EM-like algorithm becomes necessary.
> In Section A.5, we have sketched the relationship between EM (Expectation-Maximization) and MFT (Mean-Field Theory) at T>0.
> based on a similarity between the partition function in our method and the log-likelihood in the EM algorithm.
>
>
> **Efficiency of the method**
>
> We are glad you brought up the issue of computational efficiency. The equation (Eq. 3) related to optimal M does not require explicit consideration during training at T=0, and thus, our losses have no potential overhead you concerned. To clarify this point, in our revised manuscript, we have presented a pseudo code and a learning curve in runtime in the SOP dataset in Figure 4 in Appendix. This demonstrates that our methods not only converge faster in terms of epochs but also show superior performance in runtime efficiency compared to existing methods.
>
> We hope these responses and revisions adequately address your concerns and enhance the paper's value. We look forward to any further feedback you may have.

---

> > ### Comment · Reviewer_uk7M · 2023-11-21
> > **Response to authors**
> >
> > I'd like to thank the authors for their many clarifications, to me and to other reviewers.
> >
> > I am satisfied with the response and especially with the additional demonstration of run-time efficiency, a concern raised by reviewers @pubU, @aqYA, zMTp and myself.
> >
> > I am quite confident the mean-field theory is applicable to other loss functions in metric learning, and thus judge the work more positively than @q1qw, who raised a concern over that.

---

> > > ### Author Response · Authors · 2023-11-21
> > > **Thank you for your reply!**
> > >
> > > We sincerely appreciate your positive feedback and acknowledgment of our efforts in addressing the concerns raised.
> > >
> > > Your recognition of the potential applicability of mean-field theory to other loss functions in deep metric learning, is particularly encouraging.
> > >
> > > We are also grateful for the constructive dialogue that has ensued from this review process. It has undoubtedly helped in refining our manuscript and making it more comprehensive.
> > >
> > > Thank you once again for your thoughtful and detailed review.

---

### Official Review · Reviewer_aqYA · 2023-11-01

**Soundness:** 3 good
**Presentation:** 3 good
**Contribution:** 2 fair
**Rating:** 3
**Confidence:** 4

**Summary:**

This paper explores the mean field theory into metric learning  by designing two loss functions to train deep neural networks. The model's performance is evaluated on various benchmarks, including CUB, Cars, and SOP. While the paper is generally easy to follow, it lacks a sufficient level of novelty and performance improvement.

**Strengths:**

This paper explores the mean field theory into metric learning  by designing two loss functions to train deep neural networks. The model's performance is evaluated on various benchmarks, including CUB, Cars, and SOP, by comparing several other methods. The paper is generally easy to follow.

**Weaknesses:**

The major concern is that the paper lacks a sufficient level of novelty and performance improvement.

First, they mainly explore mean filed theory into metric learning. Such metric is close to central loss [R1].
[R1]. Wen, Yandong, et al. "A discriminative feature learning approach for deep face recognition." Computer Vision–ECCV 2016: 14th European Conference, Amsterdam, The Netherlands, October 11–14, 2016, Proceedings, Part VII 14. Springer International Publishing, 2016.

Second, the performance is not significant. In table I, compared with ArcFace and ProxyAnch, it is hard to justify the significant improvement. It is essential to do t-test.

**Questions:**

The clarification of model novelty.
The performance improvement.

---

> ### Author Response · Authors · 2023-11-16
> **Thank you for your review!**
>
> Thank you for your thoughtful and constructive feedback on our manuscript. We appreciate the opportunity to clarify and expand upon the points you raised.
>
> **Comparison with central loss**
>
> We acknowledge your observation regarding the similarity of our approach to central loss. However, there is a fundamental difference in methodology and computation. In central loss, there is a necessity to continually update class centers during training. In contrast, our approach rooted in mean field theory demonstrates that the optimal mean field equates to the class center. This is achieved without the extra computational cost associated with central loss. The theoretical underpinning of our method offers a more efficient alternative, emphasizing the novelty of our approach. This point is clarified in the revised manuscripts (see the last paragraph in Sec. 3.2).
>
>
>
> **Statistical significance through t-tests**
> Upon your suggestion, we conducted t-tests to validate the statistical significance of our results. The t-tests were performed comparing our methods (MFCont and MFCWMS) against ArcFace and ProxyAnchor, as shown in Tables 6 and 7 in the revised manuscript.
> The tables highlight that our MFCWMS loss demonstrates statistically significant improvements over ArcFace in all situations except 512D in the Cars dataset and ProxyAnchor except 128D and 512D in the Cars dataset. Additionally, our simpler loss function, MFCont. loss, although slightly less competitive than MFCWMS loss, still holds up well against these benchmarks in CUB and SOP.
>
> We believe these additional analyses and clarifications address your concerns regarding both the novelty and the performance improvement of our proposed methods. Our approach not only stands out in terms of theoretical efficiency over central loss but also demonstrates practical efficacy through statistically significant performance improvements.
>
> We hope that these clarifications and additional analyses sufficiently address your concerns, and we are open to any further suggestions or queries you might have.

---

> ### Author Response · Authors · 2023-11-22
> **On clarification of our previous response and updates in our revised manuscript**
>
> Thank you for your valuable feedback on our manuscript. We here clarify our previous response and notify you of the updates in our manuscript.
>
> **Distinction from central loss**
>
> We appreciate your insights regarding central loss. As detailed in our previous response, our Mean Field Theory (MFT) approach differs significantly from central loss in two key aspects:
>
> Difference in Computational Method: Central loss requires dynamic updating of class centers for regularization. In contrast, MFT achieves optimal mean fields that automatically align with class centers, eliminating the need for continual updates. This fundamental computational difference sets MFT apart from central loss.
>
> Difference in Purpose of Method: While central loss aims to enhance the performance of classification-based losses, MFT's focus is on deriving classification-based losses from pair-based ones. Our application of MFT has led to the discovery of more efficient loss functions like MFCWMS loss.
>
> **Novelty of our method**
>
> Our method's applicability to a wider range of pair-based loss functions, including those without anchors, is a substantial novelty. This flexibility allows for the discovery of more efficient loss functions.
>
> Moreover, in the revised manuscript, we have provided a detailed comparison between the proxy-based method and MFT. Theoretically, we confirmed that a loss derived from MFT more accurately approximates the original loss function, especially near the optimal solution (see Section A.4). Empirically, the MLRC benchmark results for MFNCA loss on the CUB dataset demonstrate consistent improvements in accuracy metrics (see Table 15 in Section B.7). These findings underscore the enhanced performance achieved through a more accurate approximation of the original loss functions.
>
> We believe these revisions and additions effectively address the concerns regarding novelty and comparison with both central loss and traditional proxy-based methods.
>
> We are grateful for your feedback, which has been instrumental in enhancing our manuscript. We remain open to any further suggestions or comments.

---

### Official Review · Reviewer_zMTp · 2023-11-05

**Soundness:** 3 good
**Presentation:** 2 fair
**Contribution:** 2 fair
**Rating:** 5
**Confidence:** 3

**Summary:**

This paper proposes new DML losses that are inspired from the Mean-Field Theory (MFT), which is a concept from statistical physics. Specifically, the authors follow the constructive loss and the multi-class loss to implement two MFT losses. Their extensive experiments demonstrate the efficiency of the new losses on popular DML benchmarks.

**Strengths:**

The idea of introducing unique losses based on the theory of statistical physics looks interesting and novel. No prior research has taken on this particular task.

The proposed method is evaluated on several popular DML benchmarks. The authors evaluate their method on advanced MLRC metrics, making their results convincible.

**Weaknesses:**

My major concern is that the proposed theory does not seems to be solid when it is applied on DML task. There is not enough theoretical clue that the mean-field theory (MFT) would directly benefit the DML task compared with the proxy-based losses. The authors should provide more analysis to explain the intrinsic connection between the interaction between the magnetic spin and the similarity (distance) between the data points in DML task.

The relation and comparison between the proposed loss and proxy-based loss is still not clear. The intuition behind the MFT looks similar to the proxy-based loss, where they both compare a sample with an anchor instead of all class members. Thus, a systematic compare between it and other close related losses should be provided.


It seems the computation complexity to get the mean field in Eq.5 is higher than compared with the proxies. Thus, a comparison of performance and running time may be essential to be discussed. To further clarify its arithmetic progress and complexity, it is also suggested to list the pseudo-code of the loss.


Experimental results show that the improvement in some datasets is not significant (as illustrated in table 2, figure 2). But this is not a big issue for me.

**Questions:**

Is there any assumption or internal connection between the point distances and the spin configuration?

Please respond to the above weakness.

---

> ### Author Response · Authors · 2023-11-16
> **Thank you for your review!**
>
> Thank you for your detailed and constructive feedback on our manuscript. We appreciate the time you have taken to review our work and your insightful comments, which have helped us improve the quality and clarity of our research. Below, we address each of your concerns:
>
> **Internal connection between the point distances and the spin configuration**
>
> Magnetic spin is a special case of DML, where embedding dimensions equal to 3, and $F_\theta(x)$ is L2 normalized
> This is because we introduced the magnetic spin as "a vector living on a sphere" in Sec. 3.1, and the distance is given by $d(F_\theta(x_i),F_\theta(x_j)) = 1 - F_\theta(x_i)^T F_\theta(x_i)$ when $F_\theta(x)$ is L2 normalized.
> Since DML loss is written as a function of cosine similarities in this situation, Contrastive loss reduces to the Hamiltonian of magnetic spins when $|\mathcal{C}|=1$, $m_p<0$, embedding dimension is 3, and $F_\theta(x)$ is L2 normalized, as mentioned in Sec. 3.2.
> We think these points are enough to motivate us to apply the MFT to DML loss functions.
>
> **Comparison with proxy-based approach**
> To clarify the relationship between our proposed MFT existing proxy-based approaches, we have applied MFT to a loss with an anchor in Section A.4 of the appendix. The resulting loss function has not only a term obtained in proxy-based approach (i.e., positive and negative samples are replaced by proxies) but also a term where anchors are replaced by mean fields. This term is analogous to the ProxyAnchor loss, which was previously introduced heuristically. We believe this comparison clarifies the relation and distinction between the two methodologies.
>
> **Computational complexity and pseudo-code**
>
> We understand your concern regarding the computational complexity of calculating the mean field in Eq.5. To address this, we clarify in the revised manuscript that Eq.5 represents an optimal condition of the mean field, which is not computed during training. This means the computational complexity of our approach is on par with classification-based losses. To further clarify this aspect, we have added Pseudocode in Section A.5 of the appendix, which delineates the computational process of our method.
> Additionally, in response to your suggestion, we have included a new figure (Fig. 4 in Appendix) comparing the runtimes of the mean field variants with their regular counterparts. This empirical evidence demonstrates that the mean field losses also exhibit faster runtime performance, reinforcing the practical applicability of our method.
>
>
> In conclusion, we hope that our revisions and additional explanations adequately address your concerns and enhance the clarity and impact of our work. We are grateful for the opportunity to improve our manuscript based on your feedback and look forward to any further suggestions you may have.

---

> > ### Comment · Reviewer_zMTp · 2023-11-20
> > **Thanks for your response**
> >
> > Dear authors,
> >
> > Thanks for your response. The response resolved my concern about computational complexity. However, I was still not convinced by the first two concerns. Since this may be the first work to bring concepts and theory in statistical physics into machine learning or deep metric learning, it is crucial to construct a clear explanation of the internal connection between them. If you assume magnetic spin as a special case of DML where embedding dimensions are equal to 3, what are their physical meaning when calculating the dot production (cosine similarity) between two magnetic pins? Why can we bring the Hamiltonian into the DML loss?
> >
> > Also, as proposed by reviewer aqYA, the comparison between MFT and proxy-based loss or central-based loss in the machine learning field is not clear. It is still not clear what is the benefit of MFT except the low computational cost. Why it improve the overall performance?
> >
> >
> > Best,
> > Reviewer

---

> > > ### Author Response · Authors · 2023-11-21
> > > **Thank you for your reply.**
> > >
> > > Thank you once again for your valuable feedback and for providing us with an opportunity to further clarify and improve our manuscript. Below, we address the concerns and comments:
> > >
> > > **Physics behind the Hamiltonian**
> > >
> > > The Hamiltonian (or energy) of a magnetic moment $\mathbf{m}$ in an applied magnetic field $\mathbf{B}$ is typically given by $H = - \mathbf{m}\cdot \mathbf{B}$ (e.g., Zeeman effect).
> > >
> > > In the context of ferromagnet, the each spin $\mathbf{S}_i$ (i.e., $\mathbf{m}$) interacts with the magnetic field produced by the other spins $J\sum_j\mathbf{S}_j$ (i.e., $\mathbf{B}$). As a result, each pair of spins interacts in a cosine similarity, which is parallel to the deep metric learning (DML) task where the similarity (or distance) between data points is a central concern.
> > >
> > > This discussion is added in the footnote2 and helps a reader not familiar with physics.
> > >
> > > **Comparison with proxy-based approach**
> > >
> > > Compared with the proxy-based method, we think the MFT offers two distinct advantages over traditional proxy-based methods:
> > >
> > > (1) Applicability to a wider range of loss functions:.
> > >
> > > Traditional proxy-based methods assume the dependency of loss functions on anchors, positive samples, and negative samples. This assumption limits their applicability to scenarios where an anchor is explicitly defined (e.g., they cannot be applied to loss functions like contrastive loss or CWMS).
> > >
> > > In contrast, MFT is based on the Taylor expansion of $(\mathbf{M}-\mathbf{F}_\theta(x_i))$, making it versatile and applicable regardless of whether an anchor is present. This broader applicability allows MFT to be integrated into a wider type of loss functions.
> > >
> > > (2) Better approximation of original loss functions:
> > >
> > > As elaborated in Sec. A.4, applying MFT to anchor loss generates terms that cannot be derived through proxy-based methods. These terms arise from the expansion of positive/negative samples, which are overlooked in proxy-based approaches.
> > > By considering these additional terms, MFT can approximate the original loss function more accurately, especially near the optimal solution. This point is stated in Sec. A.4 in the latest version of our manuscript.
> > >
> > >
> > > Moreover, we run an MLRC benchmark for MFNCA loss (mean field counterpart of the ProxyNCA derived in Eq. (21)) in the CUB dataset and confirm that MFNCA improves the accuracy metrics consistently (Table 15 in Sec. B.7). Thus, the better approximation of original loss functions is likely a key factor in the enhanced overall performance observed in our experiments.
> > >
> > >
> > > Regarding the comparison with central loss, we believe such a comparison is not necessary in our context. Central loss dynamically calculates and updates the class centers, employing them for regularization of mean fields and proxies. Since it is applicable to all classification-based loss functions, our focus on MFT and its advantages does not directly intersect with the realm of central loss.
> > >
> > >
> > >
> > >
> > > We hope this response sufficiently addresses the points raised and further demonstrates the novelty and effectiveness of our approach. Your feedback has been instrumental in refining our manuscript, and we remain open to any further suggestions or comments.

---

### Meta-Review · Area_Chair_s8C9 · 2023-12-06

**Metareview:**

The current paper uses mean-field theory (MFT) to approximate pair-wise losses in deep metric learning, which is based on a Taylor expansion of the corresponding loss wrt fluctuations around the mean field. The authors instantiate the method using two losses (Contrastive loss and Multisimilarity loss). Then it is tested on image-retrieval benchmark datasets, CUB, Cars, SOP, showing improved efficiency and sometimes improved accuracy.

Strengths:

- The authors' approach presents an interesting connection between metric learning with MFT (mean-field theory) in statistical physics and gives a universal tool that can be generalized to different losses. It also presents a new application of MFT in machine learning.

- The experiments are well organized and presented. On various benchmarks, the proposed approach shows better performance in terms of learning curves and the converged embedding.

Weaknesses:

As concerned by the reviewers, there should be a more delicate comparison between MFT and proxy-based losses. The authors made a comparison in Appendix A.4 and found an additional term of the MFT variants. This should contain more details with the main result explained in the paper sections. Moreover, it should be systematically compared in the experiments, e.g. baselines without the additional term.

This usefulness of the proposed approach is supported by two pairwise losses. It is not clear in theory why the mean filed method sometimes performs better than the original pairwise loss. As consistency is discussed in A.2, the authors are suggested to remark on the consistency of the two MFT losses.

It may be worth mentioning other MFT applications, in machine learning, e.g. Boltzmann machines, and other types of pair-wise losses, e.g. in manifold learning.

**Justification For Why Not Higher Score:**

Based on the reviews, there should be a technical discussion in the main text to compare clearly with the proxy-based losses. Without it, the technical part is too brief and the reader may be confused about the novelty of the proposed new family of approximation losses.

**Justification For Why Not Lower Score:**

This work introduces a universal tool into the realm of deep metric learning that can be generalized, as pair-wise losses appear in other areas as well. It is potentially interesting to the community. The writing quality is of publication standard.

---

### Decision · Program_Chairs · 2024-01-16

Accept (poster)